



# Using Eddy Covariance to Measure the Dependence of Air-Sea $CO_2$ Exchange Rate on Friction Velocity

Sebastian Landwehr[1], Scott D. Miller[2], Murray J. Smith[3], Thomas G. Bell[4], Eric S. Saltzman[5], and Brian Ward[1]

[1]School of Physics and Ryan Institute, National University of Ireland Galway, Galway, Ireland.
[2]Atmospheric Sciences Research Center, University at Albany, State University of New York, Albany, New York, USA.
[3]National Institute of Water and Atmospheric Research (NIWA), Private Bag 14-901 Kilbirnie, Wellington, New Zealand.
[4]Plymouth Marine Laboratory, Prospect Place, The Hoe, Plymouth, PL1 3DH, United Kingdom.
[5]Earth System Science, University of California, Irvine, California, USA

*Correspondence to:* Brian Ward (bward@nuigalway.ie)

**Abstract.** Parameterisation of the air-sea gas transfer velocity of $CO_2$ and other trace gases under open-ocean conditions has been a focus of air-sea interaction research and is required for accurately determining ocean carbon uptake. Ships are the most widely used platform for air-sea flux measurements but the quality of the data can be compromised by airflow distortion and sensor cross-sensitivity effects. Recent improvements in the understanding of these effects have led to enhanced corrections to

5   the shipboard eddy covariance (EC) measurements.

Here we present a revised analysis of eddy covariance measurements of air-sea $CO_2$ and momentum fluxes from the Southern Ocean Surface Ocean Aerosol Production study (SOAP). We show that it is possible to significantly reduce the scatter in the EC data and achieve consistency between measurements taken on-station and with the ship underway. The gas transfer velocities from the EC measurements correlate better with the EC friction velocity ($u_*$) than with mean wind speeds derived

10  from shipboard measurements corrected with an airflow distortion model. For the observed range of wind speeds ($u_{10N}$=3–23 m s$^{-1}$), the transfer velocities can be parameterised with a linear fit to $u_*$. The SOAP data are compared to previous gas transfer parameterisations using $u_{10N}$ computed from the EC friction velocity with the drag coefficient from the COARE model. The SOAP results are consistent with previous gas transfer studies, but at high wind speeds they do not support the sharp increase in gas transfer associated with bubble-mediated transfer predicted by physically based models.



## 1  Introduction

Mass exchange across the air-sea interface is an important component of the Earth's climate system. Uptake by the world oceans has removed approximately 25% of the anthropogenic carbon dioxide ($CO_2$) emissions from the atmosphere (Le Quéré et al., 2015). Understanding the processes that control the ocean/atmosphere exchange of $CO_2$ is important in order to estimate global carbon fluxes and to assess the evolution and future impact of ocean uptake on Earth's climate.

The flux of $CO_2$ across the air-sea interface can be written as

$$F_{CO_2} = \Delta p CO_2 \, \alpha_{CO_2} \, k_{CO_2}, \tag{1}$$

where $\Delta p CO_2$, $\alpha_{CO_2}$, and $k_{CO_2}$ are the partial pressure difference, the solubility, and the transfer velocity. The gas transfer velocity is often parameterised as polynomial function of the mean wind speed at a height of 10 meter above sea level ($u_{10N}$). Several different experimental approaches have been used to quantify air-sea gas exchange: (i) tracer studies utilising ambient gases ($^{14}CO_2$) (e.g. Wanninkhof, 1992; Sweeney et al., 2007) which integrate the flux over time scales of years, (ii) deliberately introduced tracers ($^3$He/SF$_6$) (e.g. Nightingale et al., 2000; Ho et al., 2006) which integrate the flux over time scales of days, and (iii) direct eddy covariance (EC) flux measurements on hourly time scales (e.g. McGillis et al., 2001, 2004; Kondo and Osamu, 2007; Miller et al., 2009, 2010; Prytherch et al., 2010; Edson et al., 2011; Blomquist et al., 2014). Some EC measurements tend to support a cubic wind-speed dependence for $k_{CO_2}$, (e.g. McGillis et al., 2001; Prytherch et al., 2010; Edson et al., 2011) whereas results from tracer studies (e.g. Nightingale et al., 2000; Ho et al., 2011) and more recent EC studies (Miller et al., 2010; Butterworth and Miller, 2016a) are better fitted by a quadratic model. Gas exchange measurements at high wind speeds are rare and the extrapolation of the $k_{CO_2}$ versus $u_{10N}$ relation leads to large uncertainties in global $CO_2$ uptake e.g. Takahashi et al. (2002) found a 70% enhancement in annual $CO_2$ uptake when comparing cubic to quadratic wind speed parameterisations.

Eddy covariance (EC) is a method for the direct measurement of surface fluxes of momentum, heat, or trace gases at a height of a few metres above the surface (Kaimal and Finnigan, 1994). The $CO_2$ flux is defined as the covariance of the $CO_2$ mixing ratio ($x'_{CO_2}$) with the vertical wind speed ($w'$) multiplied by the dry air density ($n_{air}$):

$$F_{CO_2} = n_{air} \left\langle w' x'_{CO_2} \right\rangle \tag{2}$$

There are several challenges associated with the shipboard use of EC to measure air-sea fluxes. One is the need to correct measured winds due to anemometer accelerations and changes in orientation due to ship motion. Another is the disturbance of the wind field by the ship (here termed air-flow distortion or AFD). Air flow over the bow and the presence of the ship superstructure can lead to deflection of the wind vector and acceleration or deceleration of the mean wind speed. Uplift of air as it passes over the ship also leads to a discrepancy between the measurement height and the height from which the sampled air originated. This can lead to biased results when Monin-Obukhov Similarity Theory (MOST) profiles are used to extrapolate shipboard measurements to a specific reference height (typically 10 meter). The resulting errors in the wind speed measurements are sensitive to the sensor location and the orientation of the ship with respect to the wind field.





Typical approaches to deal with this problem are by careful anemometer placement, by restricting the use of data to a narrow sector of relative wind direction, or by using numerical air-flow models to quantify air flow disturbance (Yelland et al., 1998, 2002; Popinet et al., 2004; O'Sullivan et al., 2013; O'Sullivan et al., 2015).

Direct comparisons between ship-board momentum flux measurements and those from low profile buoys or floating platforms (FLIP) have shown significant differences (Pedreros et al., 2003; Edson et al., 1998). Landwehr et al. (2015) showed that such discrepancies could be explained by inappropriate application of the platform motion correction and rotation of the wind vector, which led to overestimates of the deflection of the apparent wind vector by the ship's structure and provided an adapted correction.

Another concern is air flow generated by the moving platform that is not accounted by tracking the motion of the measurement volume. Ship motion is essentially wave driven and this signal can therefore manifest itself as a residual motion peak in the flux spectra. Flügge et al. (2016) showed that residual motion-correlated signals in the momentum flux spectra measured from a discuss-buoy were related to the platform motion as they were not observed in the spectra measured at a nearby tower. Prytherch et al. (2015) provided evidence that the residual motion signal in momentum flux spectra obtained on board the RRS *James Clark Ross* were caused by motion-induced flow distortion rather than by wave-induced momentum flux. They also provided a simple correction for the induced bias via linear regression of the motion corrected wind speed signal with the vertical acceleration and velocity signals. Similar methods were previously employed successfully by Yang et al. (2013).

The non-dispersive infrared $CO_2$ gas analysers used in most EC studies have cross-sensitivities to water vapour, which lead to large uncertainties in the measurements and unrealistic transfer velocity estimates (Kondo and Osamu, 2007; Prytherch et al., 2010; Edson et al., 2011; Blomquist et al., 2014). The cross-sensitivity effect can be mitigated by the use of closed-path systems in combination with a dryer to remove water vapour fluctuations in the measurement volume (Miller et al., 2010). $CO_2$ gas analysers also exhibit motion sensitivity (Miller et al., 2010), yielding signals that may covary with motion-induced apparent winds. If not fully corrected, such signals would lead to spurious fluxes.

Here we discuss the analysis of EC measurements of momentum and $CO_2$ fluxes taken on board the R/V-*Tangaroa* during the Southern Ocean Aerosol Production study (SOAP), which was conducted from February to March 2012 on the R/V *Tangaroa* (Law et al., 2017). The SOAP study was conducted in biologically productive waters on the Chatham Rise east of New Zealand. The $CO_2$ flux measurements were previously published in Landwehr et al. (2014). Here the data are reanalysed using the corrections proposed by Landwehr et al. (2015) and Prytherch et al. (2015). We describe the correction methods and discuss the resulting improvements in the quality of the EC fluxes and mean wind speeds. Air-sea gas transfer velocities are calculated using continuous underway measurements of seawater and atmospheric $CO_2$, and compared with results from previous gas-exchange studies.



## 2 Methods

### 2.1 Sea water, atmospheric and flux measurements

The EC system consisted of two Csat 3 sonic anemometers mounted on the bow mast at a nominal height of 12.6 meter above sea level (m a.s.l.). The two anemometers (port and starboard) were mounted 0.38 m away from the ship's main axis so that the distance between the two sensing volumes was 0.76 meter. An inertial motion sensor (IMU - Systron Donner MotionPak II) measured linear accelerations and angular rates along 3 orthogonal axes. The motion sensor was located between and slightly aft of the sonic anemometers. Together with a GPS compass and the ship's gyrocompass these data were used to completely describe the ship's motion following Miller et al. (2008). Two Licor non-dispersive infra-red gas analysers (IRGA) of the model LI-7500 were installed in a laboratory van on the foredeck and supplied with sample air via heated stainless steel tubing (ID = 1 cm, L = 20 m). A bypass flow system was used to provide a high flow rate of 100 standard liter per minute (slpm) through the long tubing of which a fraction (18 slpm) was passed through the gas analysers. The sample air was dried prior to analysis using a Nafion membrane dryer. The bypass flow system allowed for a high flow rate through the long sample tubing to minimize delay and loss of turbulent fluctuations. There was a pressure drop of 260 mbar between the inlet and the gas analyser. Further details on the EC system can be found in Landwehr et al. (2014).

Surface water $pCO_2$ was measured using a showerhead equilibrator based system followed by a drier and infrared gas analyser (Licor 6251). Seawater was supplied from a 5 meter depth intake through the ship's scientific seawater supply. The gas analyser was calibrated using 4 gas standards ranging in concentration from 0.0 to 406.8 ppmv. The precision of the system is estimated to be about $\pm 1\%$ (Currie et al., 2011).

Wind speed measurements from the Automated Weather Station (AWS) positioned above the crowsnest of the R/V-*Tangaroa* (25.6 m a.s.l.) are also used in this study.

### 2.2 Simulation-based air flow distortion correction

The Gerris computational fluid dynamics (CFD) model was used to simulate flow over the R/V-*Tangaroa* for a range of azimuths at 15° intervals. Gerris achieves a high degree of numerical efficiency by using an adaptive grid, increasing the grid resolution in regions of high turbulence (Popinet, 2003). The adaptive grid was limited to the 0.5 meter resolution of the numerical CAD model of the ship, which did not include details such as the foremast or handrails, nor the two vans placed on the foredeck during SOAP. The large-eddy simulations (LES) did not explicitly include viscous terms but includes "numerical viscosity" associated with the discretisation provided for subgrid-scale dissipation (Popinet et al., 2004). The inflow velocity was uniform with height, rather than a more realistic logarithmic profile. The Gerris model is started with initial conditions, the flow speed and uplift predictions were obtained from a time average of the modelled time-evolving three-dimensional turbulence, after the conditions have reached steady state. The simulations were used to estimate corrections for acceleration/deceleration and uplift of apparent wind at the Automated Weather Station (AWS) anemometer at the crowsnest (25.6 m a.s.l.) and the two anemometers at the bow mast (12.6 m a.s.l). The wind speeds were subsequently corrected for platform motion and converted to $u_{10N}$.





## 2.3 Correction for platform motion and flow distortion

In Landwehr et al. (2014), the measured wind speed was fully motion corrected before the mean tilt was estimated. This leads to an overestimation of the vertical tilt $\theta$ that scales with the ratio of the apparent and true wind speed. Here we estimate the vertical tilt angle by rotating the apparent wind vector for each 12 minute interval and subsequently applying the radial planar fit (rPF) following Landwehr et al. (2015). The vertical tilt of the wind vector varied from about $5°$ for beam on wind directions

to a maximum of $12.4°$ for bow on wind directions.

The tilt in the wind vector indicates an uplift of the air passing over the ship. For the bow mast anemometers, an uplift ranging from $0.5\,\mathrm{m}$ to $4\,\mathrm{m}$ was estimated from the observed momentum flux cospectra and from LES simulations, as described in Appendix A. The estimated undisturbed height of the sampled air ($\tilde{z}$) was used to normalise the wind speed measurements to a nominal height of $10\,\mathrm{m}$ a.s.l. On average this resulted in $u_{10\mathrm{N}}$ estimates about 2% higher than those based on the sampling

height.

## 2.4 Regression of the vertical wind speed signal with platform motion signals

Figure 1 shows average cospectra of the turbulent component of the vertical velocity with the longitudinal (alongwind) component of horizontal velocity ($n\mathrm{Co}_{\mathrm{uw}}$) during a 220 minute long period when the ship was pointed into the wind. The cospectra are shown with different levels of vertical tilt and sensor motion corrections applied.

In this example the platform motion leads to a large negative peak in the cospectrum. This was mostly removed when the measured speed was corrected for platform motion following Miller et al. (2008). However some residual structures remain in the frequency band of the ships motion $0.07\,\mathrm{Hz} \leq n \leq 0.3\,\mathrm{Hz}$. For this dataset the structure typically consisted of two peaks in opposite directions i.e. one added energy to the observed momentum flux and the other removed energy.

Prytherch et al. (2015) showed that the structures in the cospectrum are a measurement error related to the wave-induced

platform motion and suggested a regression of the wind speeds with the platform's acceleration and velocity signals to remove the erroneous signal. This motion scale correction (MSC) was used with a small modification. The acceleration and velocity signals, used in the MSC, were separated into high and low frequency components using a complementary filter at $f_c$=0.1 Hz. This procedure provided a much higher effectiveness of the MSC. Our interpretation is that the motion-scale flow distortion effects may function differently for different frequencies and types of platform motion.

It was noted that increased energy in the momentum cospectrum at low frequencies ($\leq 0.01\,\mathrm{Hz}$) was associated with small changes in ship heading and/or speed. This is presumably due to atmospheric turbulence induced by changes in ship motion. A linear regression of the vertical wind speed signal $w$ with the ships speed and heading signal (NAV - regression) was used to remove this component of the vertical wind, significantly reducing the sensitivity of the momentum flux to changes in the ship's speed and heading (Fig. 1).





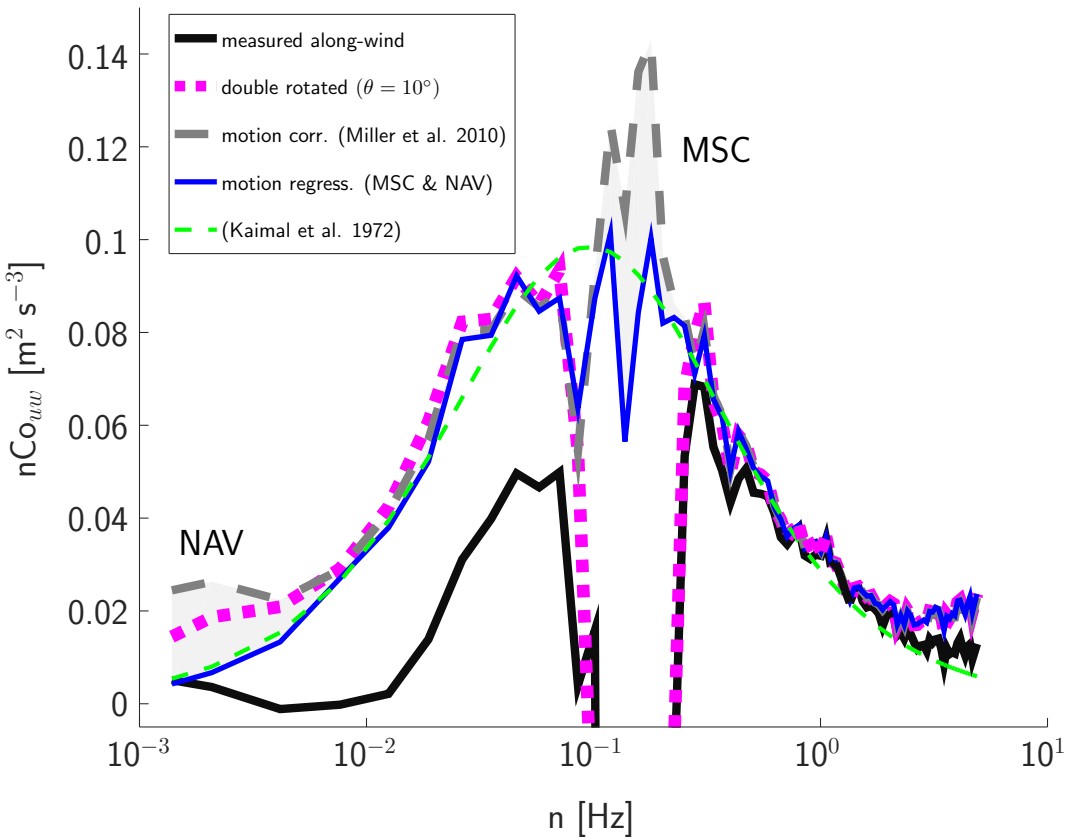

**Figure 1.** Average along-wind momentum flux cospectra (220 minute long period, relative wind direction $\alpha = 15.5°$, $u_* = 0.62\,\mathrm{m\,s^{-2}}$). Shown are spectra for different tilt-motion corrections (i) the measured wind speed corrected only for instantaneous platform orientation and wind direction ($u_\mathrm{me}$) (black line); (ii) corrected for the vertical tilt of the stream line ($\theta = 10°$) (dashed magenta line); (iii) the tilt-motion corrected wind speed (Landwehr et al., 2015) (dashed grey line); (iv) additional application of the motion scale correction (MSC), which was adapted from Prytherch et al. (2015) and regression with speed and heading (NAV), (thin blue line); (v) also shown is the semiempirical shape of the cospectrum (Kaimal et al., 1972) (green dashed line). The shaded area marks the part of the momentum-flux signal that was removed by the MSC and NAV regressions. In this case the reduction was 5% and 3% of the $u_*$ estimate, respectively.





## 2.5 Length scale dependence of the streamline coordinate system

It is conventional in EC data analysis to "rotate" the anemometer signals to correct for sensor or air flow tilt, to satisfy the condition that $(\overline{v} = 0)$ and $(\overline{w} = 0)$. Figure 1 illustrates the effect of this rotation on measured winds from the SOAP cruise. For this interval, the tilt of vertical was estimated to be $(\theta = 10°)$ upward from horizontal $(\langle w_{me} \rangle \geq 0)$. Adjusting the coordinate system for this brings $nCo_{uw}(n)$ closer to the semiempirical shape ($nCo_{uw}^{K33}$, Kaimal et al., 1972). This is true however only for $n \leq 1\,\mathrm{Hz}$. For $n \approx 1\,\mathrm{Hz}$ the shape of $nCo_{uw}(n)$ is rather independent of $\theta$ and for $n \geqslant 2\,\mathrm{Hz}$ the adjustment of the natural coordinate system causes $nCo_{uw}(n)$ to diverge from $nCo_{uw}^{K33}$. At those higher frequencies $nCo_{uw}(n)$ was well matched for $\theta = 0°$. In this example $U \approx 13.5\,\mathrm{m\,s^{-1}}$, hence $n = 1\,\mathrm{Hz}$ corresponds to a length scale of 1 meter. This suggests that while rotation of the coordinate system into the air stream is crucial to adequately measure the contribution of large eddies, it is counter-productive for the measurement of flux carried by small eddies, i.e., it may be that small-scale turbulence adjusts to the new orientation of the tilted stream lines.

The elevated cospectral energy for $n \geq 1\,\mathrm{Hz}$ is only observed in $nCo_{uw}(n)$ while the heat flux spectra of the anemometers speed-of-sound temperature ($Co_{w\theta}(n)$) tend to collapse into the expected $f^{-4/3}$ shape (see Fig. B2 in the Appendix). Due to the projection of the auto-covariances of the three components $u, v$, and $w$, the momentum flux estimate is generally more sensitive to the choice of the coordinate system than the scalar fluxes. Elevated energy in $nCo_{uw}$ at high frequencies has also been observed by, e.g., (Butterworth and Miller, 2016a), who observed wind vector tilt angles of up to $15°$ with anemometers mounted on the bow mast of the R/V *Nathaniel B. Palmer* .

In order to quantify the potential bias in the momentum flux the integration of $nCo_{uw}$ was separated into the part below and above $n = 1\,\mathrm{Hz}$, where for the frequencies above $n = 1\,\mathrm{Hz}$ the observed cospectrum $nCo_{uw}(n \geq 1\,\mathrm{Hz})$ was replaced with $nCo_{uw}^{K33}(\tilde{z}, U, L)$ as predicted by (A1). The bias can then be formulated as:

$$\Delta u_*^2 = \langle uw \rangle^{-1} \int_{1\,\mathrm{Hz}}^{5\,\mathrm{Hz}} \left[ nCo_{uw} - nCo_{uw}^{K33}(\tilde{z}, U, L) \right] dn, \tag{3}$$

where $\langle uw \rangle$ is computed from the integration of $nCo_{uw}$ over the full frequency range.

The results are plotted in Fig. B3 in the Appendix. The overestimation of $u_*$ as estimated from (3) is 2% on average but ranges from 0%–6% and appears to be a function of $\frac{\tilde{z}}{U}$ and $\frac{\tilde{z}}{L}$, which define the fraction of spectral energy at $n \geq 1\,\mathrm{Hz}$. Relative wind direction is also important in determining the overestimation of $u_*$. This measurement bias could be reduced by placing the anemometer further away from the ships hull, in order to reduce the vertical tilt of the wind vector.

In order to compare the EC based measurements of the air-side friction velocity ($u_*$) with other wind speed measurements, they are converted to $u_{10N}$ using the wind speed dependent drag coefficient from COARE version 3.5 (Edson et al., 2013). This done by iterating three times through the following equations:

$$C_D = \left( \frac{\kappa}{log_{10}(10\,z_0^{-1})} \right)^2, \tag{4}$$

$$u_{10N} = \frac{u_*}{\sqrt{C_D}} \tag{5}$$



where $(z_0 = \gamma \nu u_*^{-1} + \alpha u_*^2 g^{-1})$ is the roughness length depending on, gravity $(g)$, kinematic viscosity $(\nu)$, roughness Reynolds number for smooth flow $(\gamma = 0.11)$, and the wind speed dependent Charnock parameter,

$$\alpha = \begin{cases} 0.0017\,u_{10N} - 0.005, & (u_{10N} \leq 19.4\,\mathrm{m\,s^{-1}}). \\ 0.028, & (u_{10N} > 19.4\,\mathrm{m\,s^{-1}}). \end{cases} \tag{6}$$

that is recomputed for each iteration (Edson et al., 2013).

## 2.6 Regression corrections applied to the $CO_2$ signal

LI-7500-measured $CO_2$ densities were converted to mixing-ratios using the simultaneously measured pressure, temperature and water vapour density in the measurement volume. The LI-7500 deployed in this experiment have sensitivity to motion (Miller et al., 2010). Following Miller et al. (2010) the residual motion signal was quantified for each 12 minute interval by a linear regression of the $x_{CO_2}$ signal against the three acceleration signals and subtracted from the $x_{CO_2}$ signal. The LI-7500 sensors also have cross sensitivity to $H_2O$ (Kohsiek, 2000). The Nafion dryer removed humidity fluctuations effectively, reducing the ambient $H_2O$ flux on average by 93% (Miller et al., 2010; Landwehr et al., 2014). A similar reduction was observed for the temperature flux signal due to heat exchange across the tubing walls. Small $(< 10\%)$ differences in the $CO_2$ fluxes measured by the two LI-7500 units correlated with the residual humidity and temperature "fluxes" measured by the two dry close-path IRGA units. The bias signal was quantified by linear regressions of $x_{CO_2}$ with $x_{H2O}$ and the cell Temperature $T_{cell}$ and subtracted from the $x_{CO_2}$ signal. This reduced the disagreement between the $CO_2$ fluxes measured by the two units and the scatter in the $CO_2$ flux time series significantly. Since the variations of $T_{cell}$ and $x_{H2O}$ in the CP-IRGA are fully decoupled from the atmospheric variations by the long sample tubing and the diffusion dryer, there is no danger of removing real $CO_2$ flux signal with this regression. The regression resulted in a small reduction of the observed CO2 fluxes $(< 2\%$ on average) and a 20% reduction in variability of the flux signal. The mean, median, and standard deviation of the $CO_2$ flux were -5.14, -4.85, 2.60, respectively with the regression being applied, and -5.23, -4.84, 3.07 $\mathrm{mol\,m^2 yr^{-1}}$, without regression.

## 2.7 Correction of the $CO_2$ fluxes for attenuation of high frequency fluctuations.

In order to assess the reduction of the $CO_2$ flux signal measured with the closed-path analysers due to high frequency attenuation, resulting from the long inlet tubing, the normalised $\langle wCO_2 \rangle$ cospectra were compared with the cospectra of the sonic speed of sound temperature $\langle w\theta_s \rangle$. The flux loss was estimated as the ratio of the cumulative sums counting from low to high frequencies (ogives) at $n = 0.3\,\mathrm{Hz}$ (Marandino et al., 2007; Blomquist et al., 2010). High frequency fluctuations of $CO_2$ are attenuated by the measurement system due to passage of air through the intake tubing, drier, and closed path detector. According to similarity theory, the fraction of $CO_2$ flux lost due to high frequency attenuation depends on wind speed and atmospheric stability: For high wind speeds the cospectra are shifted to higher frequencies, thus increasing the relative loss in the $CO_2$ flux. At low wind speeds stratification $(z/L > 0)$ can suppress large scale motion. In this case the spectral peak is shifted to higher frequencies, when compared to neutral or unstable atmospheric stability $(z/L \leq 0)$. For a moving observer, the apparent wind





speed is the relevant velocity scale to predict the frequency distribution of the turbulent motion that is observed by the EC system. This is illustrated in Fig. B1 in the Appendix.

The high frequency attenuation is assessed by comparing the normalised cospectra of $CO_2$ ($nCo_{w\theta}$) and sensible heat using the sonic temperature ($nCo_{w\theta}$). This is based on the assumptions that the cospectra of $CO_2$ and sensible heat are similar (Sahlée et al., 2008), that the measured $nCo_{w\theta}$ is not attenuated, and that the attenuation of the $CO_2$ flux spectrum becomes negligible

for low frequencies ($n \leq 0.3\,\mathrm{Hz}$). The last assumption was tested by using slightly higher and lower frequencies for the loss estimation. The $CO_2$ flux data were corrected by applying a gain ($g_{CO_2}$) computed as the ratio of the cumulative sums (from low to high frequencies) of sensible heat and $CO_2$ cospectra at $n = 0.3\,\mathrm{Hz}$ (Marandino et al., 2007; Blomquist et al., 2010). In Fig. 2 the gain estimates ($g_{CO_2}$) are plotted as function of the stability parameter $\zeta$. For this plot the data set was further reduced by requiring $|\Delta T| = |T_{\mathrm{air}} - T_{\mathrm{sea}}| \geq 1\,\mathrm{K}$ and $|\Delta pCO_2| < 50\,\mathrm{ppm}$. The gain ($g_{CO_2}$) for this cruise was parametrerised

as a function of relative wind speed $U$ and the stability function $\left[ H(\frac{z}{L}) \right]^{3/4}$, with $H = 6.4\zeta + 1$ taken from Kaimal et al. (1972), as follows:

$$g_{CO_2}(U, \frac{z}{L}) = A_g \cdot U + B_g \cdot \left[ H(\frac{z}{L}) \right]^{3/4} + C_g \tag{7}$$

where $A_g = 0.0038(\pm 0.0026)[\mathrm{m\,s^{-1}}]^{-1}$, $B_g = 0.37(\pm 0.06)$, and $C_g = 0.61(\pm 0.07)$. This function was used to correct the measured $CO_2$ fluxes.

On average the correction to the $CO_2$ flux signal was found to be 4%. For the range of wind speeds ($U \leq 25\,\mathrm{m\,s^{-1}}$) and stratification ($\frac{z}{L} \leq +0.2$) on the SOAP cruise, the effect of stratification on the signal attenuation (0–30%) is larger than the effect of the relative wind speed (0–10%). It is therefore necessary to predict the attenuation of the closed-path derived scalar fluxes based on both *apparent* wind speed *and* atmospheric stability.





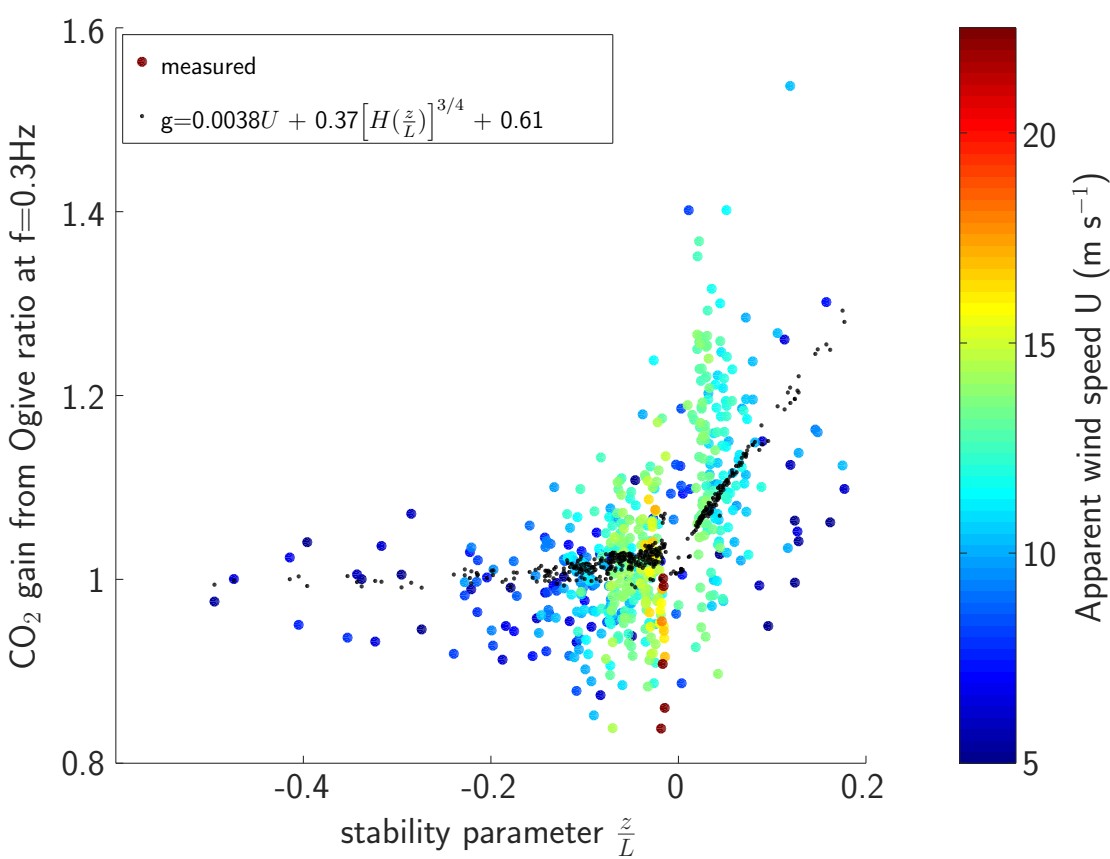

**Figure 2.** $CO_2$ gain factor ($g_{CO_2}$) as function of apparent wind speed $U$ (color) and stability parameter $\frac{z}{L}$. The result of a simple regression analysis ($g = A_g \cdot U + B_g \cdot \left[ H(\frac{z}{L}) \right]^{3/4} + C_g$) is shown as black dots.



## 2.8 Gas transfer velocity calculations

The time series of the 3D wind speed and $CO_2$ mixing ratio were separated into 12 minute periods, over which all averages and covariances were calculated. Equation (2) was used to obtain the $CO_2$ fluxes (unit $\mathrm{mol\,m^{-2}\,s^{-1}}$) which were converted to gas transfer velocities in units of $\mathrm{cm\,hr^{-1}}$:

$$k_{Sc} = (3600 * 100 * 10^6) \frac{F_{CO_2}}{\alpha_{CO_2} \Delta pCO_2},$$ (8)

where $\alpha_{CO_2} [\mathrm{mol\,m^{-3}\,atm^{-1}}]$ is solubility of $CO_2$ in sea water (Weiss, 1974).

In order to account for the influence of the sea surface temperature and salinity the transfer velocities were normalised to a Schmidt number of 660, which corresponds to $CO_2$ at 25°C

$$k_{660} = k \left( \frac{660}{Sc_{CO_2}} \right)^{-n}$$ (9)

During SOAP the Schmidt number varied between 820 and 960, for a Schmidt number exponent of $n = 1/2$, this corresponds
to a normalisation factor $(Sc/660)^{1/2}$ of 1.12 to 1.21. Laboratory studies have shown a smooth transition of $n$ from 2/3 to 1/2, when the water surface changes from smooth to rough with increasing wind speed (Jähne et al., 1984). Esters et al. (2017) showed that assuming a wind speed dependent Schmidt number can improve gas transfer velocity parameterisations. For this work, however, the choice of Schmidt number exponent has only small effect on the overall results (for $Sc = 900$ $n = 2/3$ or $n = 1/2$ corresponds to a change in the normalisation factor from 1.17 to 1.23). For simplicity $n = 1/2$ was used for the whole
dataset. Equation (9) assumes that the gas transfer velocity is purely interfacial. Bell et al. (2017), however, showed that for $u_{10N} > 10\,\mathrm{m\,s^{-1}}$ bubble mediated transfer becomes significant for the air-sea gas exchange of $CO_2$. Therefore a more complex Schmidt number/solubility normalisation may be necessary, to treat the interfacial and bubble-mediated component of the $CO_2$ gas transfer velocity separately. This is however beyond the scope of this work. Due to the small range of Schmidt number variation observed during SOAP the effect of such a normalisation on the observed wind speed dependency of $k_{CO2}$ should be
minor.

## 3 Discussion of SOAP Data Analysis

### 3.1 Effect of the tilt-motion correction on gas transfer coefficients measured underway vs on station

As described in Sec. 2.3, the eddy covariance data were analysed using the improved tilt motion correction developed by Landwehr et al. (2015). Figure 3 shows the impact of that correction on the SOAP $CO_2$ transfer velocities, $k_{660}(CO_2)$. The
improved correction has relatively little impact on transfer velocities measured while the ship was on station ($u_{ship} \leq 1\,\mathrm{m\,s^{-1}}$), giving results that are similar to those previously published by Landwehr et al. (2014). However, the transfer velocities obtained while the ship was underway ($u_{ship} > 1\,\mathrm{m\,s^1}$) were significantly reduced (up to $20\,\mathrm{cm\,hr^{-1}}$) using the improved correction. The new tilt-motion correction method eliminates the systematic bias between ship on station and ship underway data. The corresponding tilt estimates are shown in Fig. A1 in the Appendix.




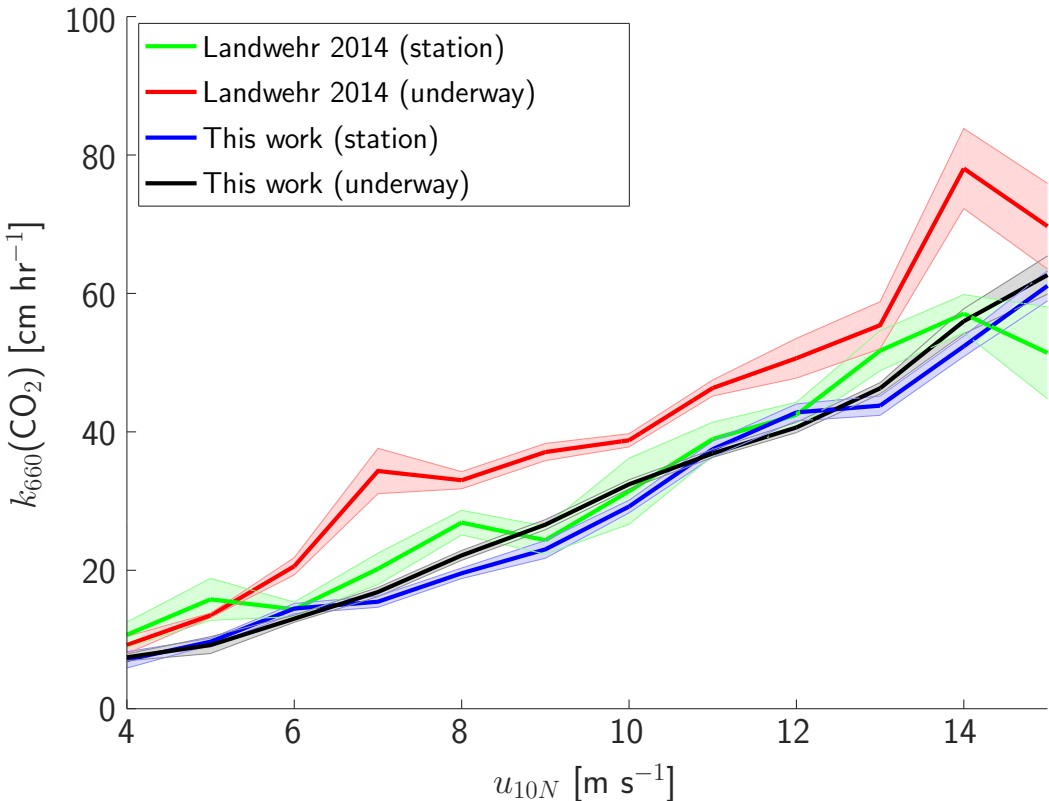

**Figure 3.** Estimated $CO_2$ transfer velocities separated into "station" ($u_{\mathrm{ship}} < 1\,\mathrm{m\,s^{-1}}$) and "underway" ($u_{\mathrm{ship}} \geq 1\,\mathrm{m\,s^1}$) and bin averaged over $1\,\mathrm{m\,s^{-1}}$ wind speed bins. Shown are the estimates from Landwehr et al. (2014) where the Double rotation of the wind vector was performed after it had been corrected for mean- and wave-motion induced ship motion (green and red); and from this work, where the radial planar fit method has been employed to estimate the vertical tilt of the wind vector (Landwehr et al., 2015). The shaded area mark one standard deviation from the bin average. The plot has been restricted to the wind speed range $4\,\mathrm{m\,s^{-1}} - 15\,\mathrm{m\,s^{-1}}$ where sufficient on station and the underway measurements were both available.





## 3.2 Effect of the air flow distortion corrections on friction velocity

Figure 4 shows the $u_*$ obtained from EC with and without the regression and high frequency correction applied as function of the air-flow distortion corrected wind speed measured at the bow mast (average of port and starboard anemometer). The corrections lead to a reduction in the friction velocity by about 10% and to a better correlation with the air-flow distortion corrected wind speed measured at the bow mast. The $u_*$ obtained from EC with and without the regression and high frequency correction are 12% and 22% higher than the $u_*$(bulk) derived from the bow mast wind speed using the COARE 3.5 (Edson et al., 2013) and a linear fit of $u_*$(EC) to $u_*$(bulk) explained 94% and 90% of the variability, respectively. The disagreement with the COARE 3.5 parameterisation is likely due to residual flow distortion errors in either the mean wind speeds or the friction velocities that were not completely removed by the applied corrections.





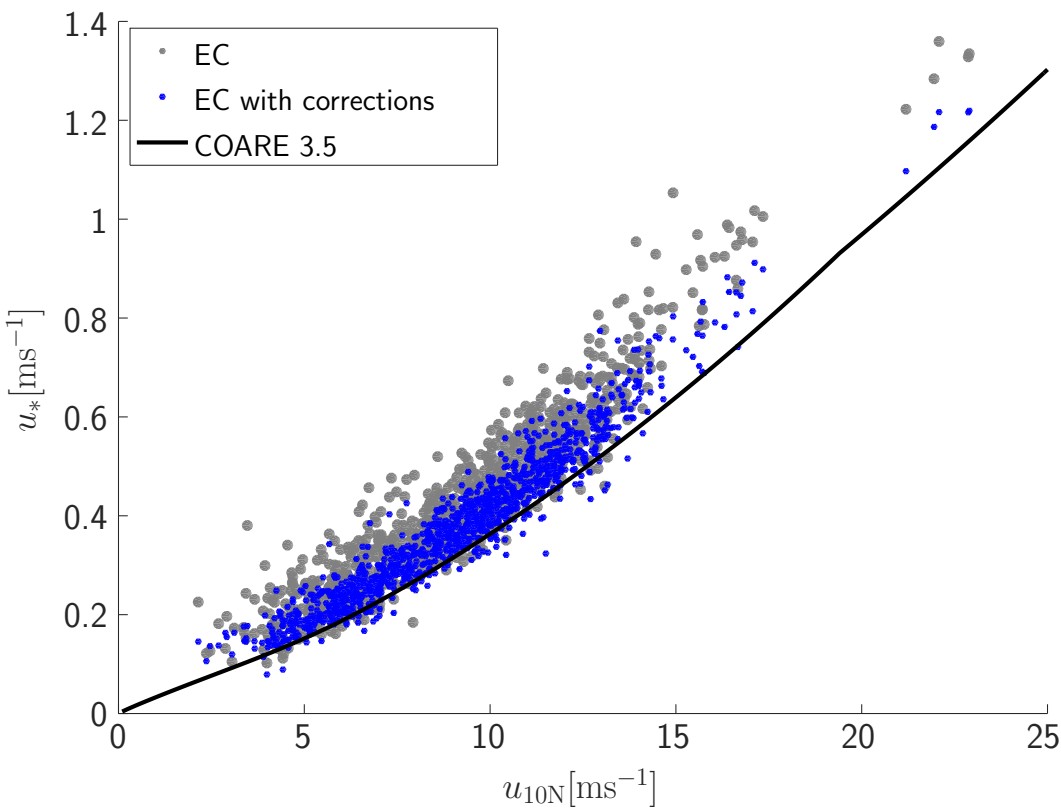

**Figure 4.** Eddy Covariance estimates of $u_*$ as function of $u_{10N}$, estimated from the air-flow distortion corrected wind speed measured by bow mast anemometers. The COARE 3.5 open ocean relation (Edson et al., 2013) is shown as black line.





### 3.3 Wind speeds measured on free floating catamaran as external reference

During periods of fair weather wind speed and direction were also measured by an Airmar sonic anemometer at 5.6 m a.s.l. on the mast of a small catamaran. These data can provide an external reference to assess the uncertainties associated with the corrected shipboard measurements. For this comparison, we use data collected when the catamaran was free floating within 10 km of the ship (i.e. not dragged by the ship or small boat), and when significant wave heights were below 2.1 meters.

The catamaran measurements were adjusted to 10 meter height neutral stability using the $u_*$ and $L$ measured with the ship-borne EC system. The same adjustment using the bulk $u_*$ and $L$ derived from the AWS wind speed measurements would result in slightly lower ($< 2\%$) $u_{10N}$ estimates. Figure 5 shows a comparison of the $u_{10N}$ estimates from the various shipborne wind measurements with those based on the catamaran. Compared to the $u_{10N}$ estimates from the catamaran wind speed measurements, the EC based results are 4% higher while the mean wind speed based estimates from the air-flow distortion corrected shipborne measurements are 8% and 15% lower for bow and crowsnest anemometer, respectively. This shows that the direct EC measurements of $u_*$ enable a better estimate of the undisturbed wind speed than the flow distortion corrected wind speeds. Note that for the adjustment of the bow mast wind speeds the estimated height of the undisturbed streamlines ($\tilde{z}$) has been used, which varies between 9-12 m.a.s.l. If instead the height of the bow anemometer ($z = 12.6\,\mathrm{m}$) was used to adjust the wind speed measurements a 2% lower $u_{10N}$ would be estimated, providing an on average 10% underestimation of the catamaran wind speeds.




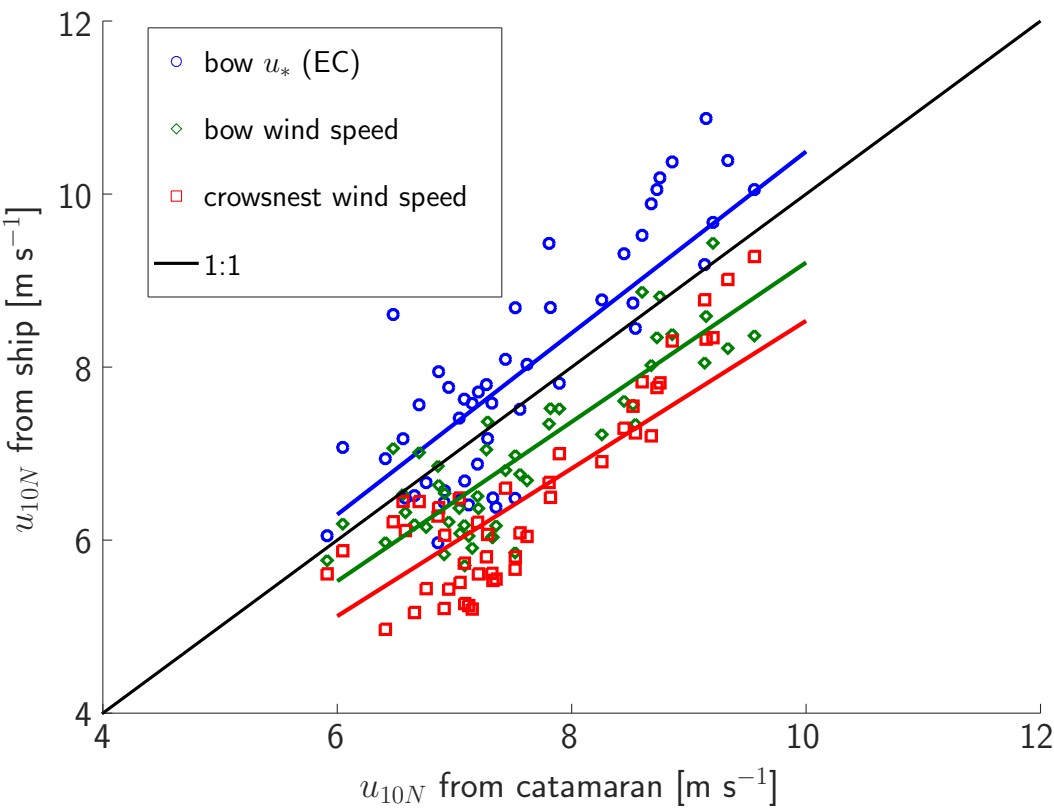

**Figure 5.** Scatter of the normalised wind speed ($u_{10N}$) measured by the Catamaran ($z_C = 5.6\,\mathrm{m}$) and the wind speed estimates from the ship: (i) from the bow mast friction velocity (converted via COARE 3.5); (ii) from the bow mast wind speed ($\tilde{z}_b = [9 - 12]\,\mathrm{m}$); and (iii) from the AWS anemometer at the crowsnest ($z_M = 25.6\,\mathrm{m}$). The lines indicate the mean ratio of the shipborne wind speeds with those from the catamaran. The ratios are $1.04(\pm 0.10)$, $0.92(\pm 0.07)$, and $0.85(\pm 0.08)$ for the bow mast momentum flux and the bow mast and crowsnest wind speeds, respectively.





### 3.4 Regression corrections and the observed correlation between wind forcing and gas transfer.

Wind forcing or wind stress ($\tau = \rho_{air} u_*$) is the major driver of near surface turbulence, and the most important parameter to predict air-sea gas exchange of $CO_2$. In order to assess the corrections that were applied to the direct flux measurements the measured $k_{660}(CO_2)$ were parametrised as a polynomial function of $u_*$. Least squares regressions of $k_{660}(CO_2)$ to $u_*$ were examined for different levels of corrections applied to the data (Table 1). Linear and quadratic fits gave equivalent goodness

of fit for the whole data set and provided very similar results for the wind speed range between $6\,\mathrm{m\,s^{-1}} \leq u_{10N} \leq 16\,\mathrm{m\,s^{-1}}$, therefore linear fits are used here.

The Deming regression (Deming, 1943) was used, which accounts for errors in observations on both the independent and dependent variable. The ratio of the relative uncertainty ($[\sigma_k/\overline{k}] : [\sigma_{u_*}/\overline{u_*}]$) was estimated from the standard deviation of the $k_{660}(CO_2)$ and $u_*$ values when these were averaged over 4 hour long periods. The ratio of relative uncertainty was ap-

proximately $6:1$ for uncorrected EC results and approximately $2:1$ for the EC results with regression corrections applied, respectively. The Deming regression resulted in slightly steeper slopes than the normal linear regression, which only accounts for uncertainty in the dependent variable ($k$).

The regression corrections applied to the $CO_2$ mixing ratios and 3D wind speed measurements significantly improve the $k_{660}(CO_2)$ to $u_*$ correlation, resulting in an increase in $R^2$ from 0.35 to 0.83. These corrections did not significantly change the

slope of $k_{660}$ vs $u_*$. The high frequency loss correction applied to the $k_{660}$ estimates and correction of $u_*$ for elevated cospectral energy at $n \geq 1\,\mathrm{Hz}$ did not improve the fit further, but slightly increased the slope of $k_{660}$ vs ($u_*$) by about 7%. Figure 6 shows a scatter plot of $k_{660}$ and $u_*$ for the uncorrected EC results and for the data with all motion regression corrections applied to $F_{CO_2}$ and $u_*$.

Table 1 also shows the results for least squares regressions of $k_{660}(CO_2)$ to $u_*$(bulk), which were derived from the wind

speeds measured at the bow mast and at the crowsnest (AWS).

The corrections applied in the data analysis are summarised in Table 2, which provides the range and mean of the relative bias in the EC friction velocities and $CO_2$ fluxes that were removed by each of the corrections. Most corrections, on average, reduced the results of the EC flux measurements. Only the correction for signal attenuation of the $CO_2$ fluxes in the long sample tubing caused an increase in the $CO_2$ fluxes.



**Table 1.** Cumulative effect of various corrections on the relationship of gas transfer and friction velocity. Slope $a$ and offset $b$ with standard error ($\pm$SE) and coefficients of determination ($R^2$) for linear Deming regression of $k_{660}(CO_2)$ with $u_*$, ($k_{660} = a\,u_* + b$) (Deming, 1943). The first five rows feature EC results with increasing level of corrections applied to the data. The two last rows show the results for using $u_*$(bulk) derived from the wind speed measurements on bow mast and crowsnest (both were corrected for air-flow distortion).

| Corrections applied (cumulative) | slope a($\pm$SE) $\left[\mathrm{cm\,hr^{-1}(m\,s^{-1})^{-1}}\right]$ | offset b($\pm$SE) $\left[\mathrm{cm\,hr^{-1}}\right]$ | $R^2$ $[-]$ |
|---|---|---|---|
| Basic EC | 114.7($\pm$6.1) | -11.2($\pm$2.2) | 0.35 |
| EC +...+ $x_{CO_2}$ regr. | 98.2($\pm$2.7) | -6.4($\pm$1.0) | 0.75 |
| EC +...+ MSC regr. | 94.9($\pm$2.4) | -6.7($\pm$0.9) | 0.77 |
| EC +...+ NAV regr. | 97.6($\pm$2.3) | -6.5($\pm$0.8) | 0.83 |
| EC +...+ high freq. corr | 104.8($\pm$2.3) | -7.3($\pm$0.8) | 0.83 |
| Bow mast wspd. | 114.3($\pm$3.0) | -5.0($\pm$0.9) | 0.78 |
| Crowsnest (AWS) wspd. | 109.4($\pm$2.4) | -3.7($\pm$0.7) | 0.76 |





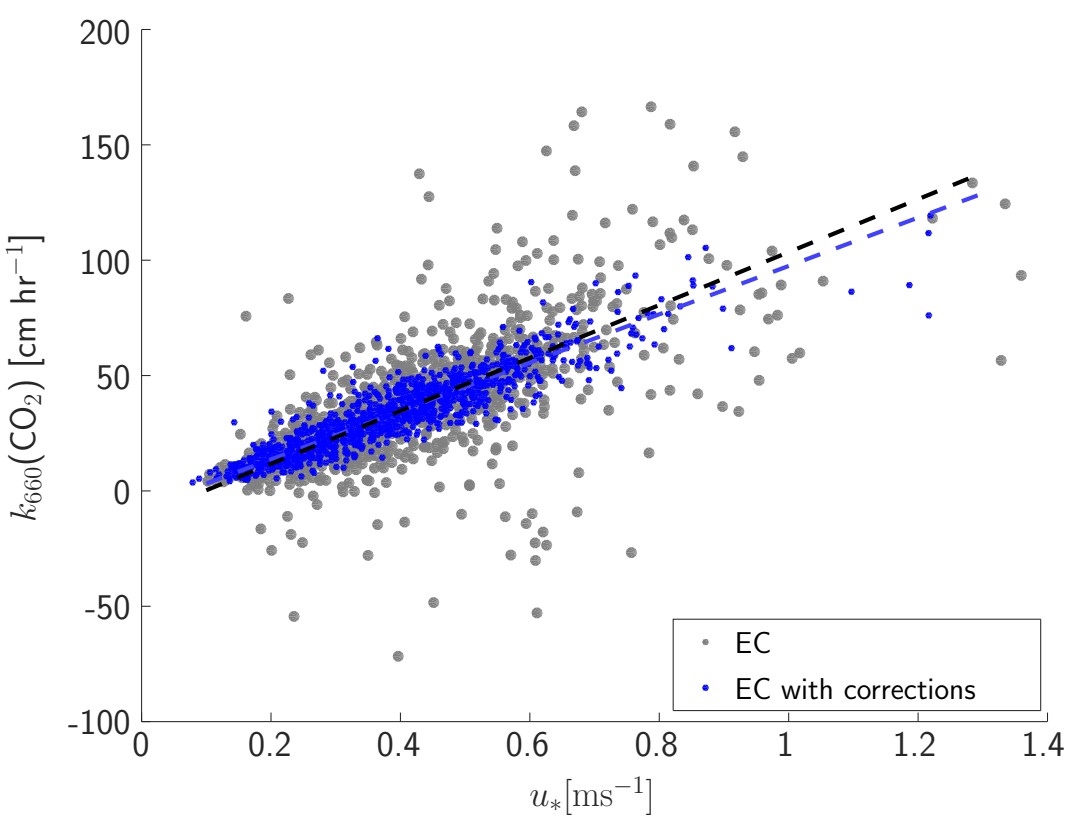

**Figure 6.** $CO_2$ gas transfer velocity normalised to $Sc = 660$ as function of $u_*$. Shown are the direct EC estimates prior and after the application of the regression corrections to the $CO_2$ mixing ratios and wind speed measurements. The lines show linear fits of $k_{660}(u_*)$ against $u_*$. The linear regression explained 83% of the EC results, respectively with the regression corrections applied and 35% without regression corrections.



**Table 2.** Corrections to the EC data listed in the order in which they were applied. Column #2 notes to which data the correction is applied. In column #3 the range of bias is given in % of the corrected value, with negative numbers indicating that the corrected value was underestimated by the uncorrected observation. The mean and standard deviation of the relative corrections for the SOAP data are provided in column #4. Column #5 provides references were applicable. $^{\dagger}$ For the motion regression applied to the $CO_2$ mixing ratios the bias ranges are given for the two cases that the MSC was or [ was not ] applied to the 3D wind speeds.

| Name (abbreviation) | Applied to | Range of relative bias | | mean($\pm$ std) of | References |
|---|---|---|---|---|---|
| Tilt-motion correction (rPF) | (u,v,w) | $u_*$: | -50% to +500% | +80($\pm$50)% | Edson et al. (1998) Landwehr et al. (2015) |
| Motion regression (MSC) | (u,v,w) | $u_*$: | -50% to +110% | +3($\pm$ 10)% | Prytherch et al. (2015) (with modifications) |
| Speed/heading regression (NAV) | (u,v,w) | $u_*$: | -30% to +120% | +5($\pm$ 9)% | this work |
| High frequency elevated energy | $nCo_{uw}$ | $u_*$: | 0% to +6% | +2($\pm$ 1)% | this work |
| $^{\dagger}$Regression with motion, $x_{H_2O}, T$ | $x_{CO_2}$ | $F_{CO_2}$: | -100% to +150% [ -200% to 300% ] | +1.5($\pm$ 20)% [+14($\pm$ 40)% ] | Miller et al. (2010) (with modifications) |
| High frequency loss correction | $nCo_{wx}$ $(x = x_{CO_2})$ | $F_{CO_2}$: | -30% to +0% | -4($\pm$ 5)% | Marandino et al. (2007) Blomquist et al. (2010) |



 ### 3.5    Correlation between transfer velocity and different wind speed estimates

In order to study how air-flow distortion influenced the relationship between $k_{660}$ and $u_*$, the relative anomaly from the fit prediction ($\delta k = k_{\text{meas.}}/k_{\text{fit}}(u_*) - 1$) was calculated and bin-averaged over 25°wind direction sectors (Fig. 7). No difference was observed when using $1^{st}$, $2^{nd}$, or $3^{rd}$ order polynomials to fit the data. For the air-flow distortion corrected wind speed from the AWS anemometer the anomalies show a strong directional pattern spanning more than 25% variation. This can be

attributed to air-flow distortion effects (including height displacement), which were not properly accounted for by the LES model. The AWS wind speed explains 77% of the variability in $k_{660}$. For the $u_*$(bulk) calculated from the bow mast wind speeds (corrected for air-flow distortion and effective measurement height) the directional variability is considerably reduced and the correlation explains 79% of the variability in $k_{660}$. The $u_*$ values derived from the eddy covariance momentum flux measurements exhibited the least directional variability in the $\delta k$ and explained 83% of the variability in $k_{660}$.

Based on these results, the $k_{660}$ measured on SOAP are best reported as function of the directly measured $u_*$ (EC). This result might apply to other ship-borne EC gas flux studies where disagreement between direct and bulk estimates of the momentum flux have been recorded. For example, on the R/V-*Knorr* bow mast, $u_*$ measured (EC) at 13.6 m a.s.l. agreed well with the COARE-predicted $u_*$ when plotted against $u_{10\text{N}}$ derived from an anemometer at 15.5 m a.s.l. When $u_{10\text{N}}$ was derived from the 13.6 m a.s.l. anemometer, the COARE-predicted substantially underestimated the measured $u_*$ (EC) across a range of wind

speeds (Bell et al., 2013, supplementary info). Scaling $k$(EC) with $u_*$ or $u_{10\text{N}}$(EC) instead of $u_{10\text{N}}$(bulk) avoids uncertainties arising from height and stability corrections as well as potential bias arising from air-flow distortion effects that might affect EC and mean wind speed measurements differently.





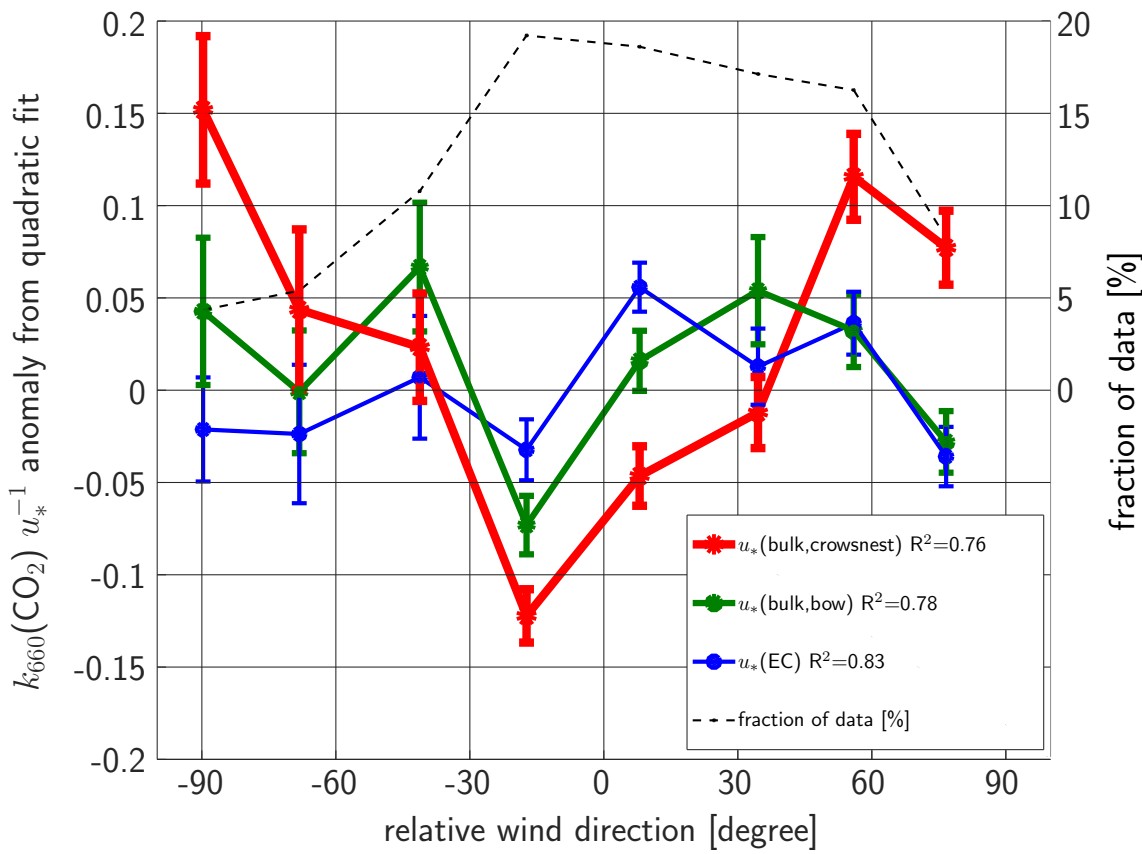

**Figure 7.** Average relative deviations ($\delta k = k_{\mathrm{meas.}}/k_{\mathrm{fit}}(u_*) - 1$) from linear fits to $k_{660}(u_*)$ as a function of the relative wind direction. The relative anomalies are averaged into relative wind direction sectors (25°bins). The errorbars indicate the standard error of the mean value of each sector. The red and green line show the anomalies for direct EC $k_{660}$ and bulk $u_*$ derived from the AWS (crowsnest) and the bow mast anemometer respectively. The blue line shows the anomalies for the ratio of direct EC $k_{660}$ and direct EC $u_*$. The coefficients of determination ($R^2$) of the polynomial fits are provided in the legend. The black dashed line indicates the fraction of data observed in each wind direction sector.



## 4 Discussion of SOAP Gas Transfer Velocities

### 4.1 SOAP gas transfer velocity as a function of friction velocity

The SOAP data set consists of 1155 measurements (231 hours), ranging in wind speed ($u_{10N}$) from $3 - 23\,\mathrm{m\,s^{-1}}$, with the majority of the data (95%) between $5 - 16\,\mathrm{m\,s^{-1}}$. The SOAP gas transfer velocities are highly correlated with wind forcing and wind speed. For the observed range of wind speeds the relationship to friction velocity is well described by a linear fit to the measured (EC) $u_*$ (Fig. 8).

$$k_{660} = 104.8(\pm 2.3)\,u_* - 7.3(\pm 0.8), \tag{10}$$

where the units of $k_{660}$ and $u_*$ are $[\mathrm{cm\,hr^{-1}}]$ and $[\mathrm{m\,s^{-1}}]$, respectively. The fit explains 83% of the observed variability in the gas transfer velocity. As noted in section 2.6, regressions with higher order polynomials do not improve the fit. The SOAP gas transfer velocities exhibit much less scatter as function of wind speed than previous $CO_2$ EC flux studies (McGillis et al., 2001; Edson et al., 2011), providing for a more precise estimate of the wind speed dependence.

Extrapolation of this linear $k$ vs EC $u_*$ relationship outside of the wind speed range of the SOAP data set is not recommended, because there are physical reasons why this relationship might not hold. At lower wind speeds, buoyancy-driven processes may contribute significantly to gas transfer (Soloviev, 2007). In fact (10) slightly underestimates the wind speed binned data for $u_{10N} < 5\,\mathrm{m\,s^{-1}}$) and would predict negative $k_{660}$ for $u_* \le 0.07\,\mathrm{m\,s^{-1}}$ ($u_{10N} \le 2.3\,\mathrm{m\,s^{-1}}$)). At high wind speeds, wave breaking and bubble-driven gas transfer are expected to contribute to gas transfer of $CO_2$ and other sparingly soluble gases (Woolf, 1997; Fairall et al., 2011; Bell et al., 2017). Surprisingly, there is no evidence in the SOAP data for an increase in the slope of the $k_{660}$ vs $u_*$ relationship at high wind speeds. If anything, the limited SOAP data available at the highest wind speeds appear to be biased low relative to the linear regression.



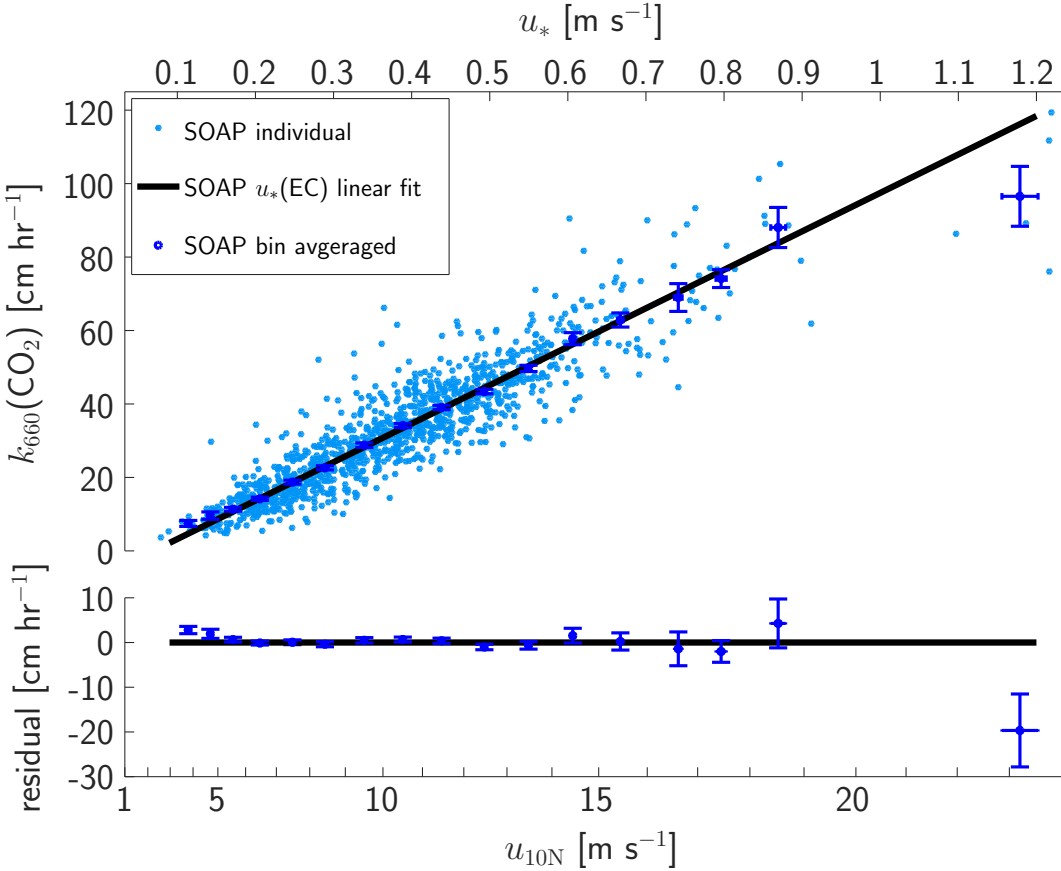

**Figure 8.** Top: $CO_2$ gas transfer velocity normalised to $Sc = 660$ as function of $u_*$. The data are shown as individual measurements (light blue) and bin averaged over $1\,\mathrm{m\,s^{-1}}$ wind speed bins (dark blue) (the width of the last two bins was increased to $3\,\mathrm{m\,s^{-1}}$, in order to account for the scarcity of the data at high wind speeds). The error bars indicate the standard deviation. The linear regression to the individual data (10) is shown as black line. Bottom: residual difference of the bin averages from (10). The two x-axis show the friction velocity and the corresponding $u_{10N}$ when the COARE 3.5 drag coefficient is assumed.





## 4.2 Comparison of SOAP results to previous gas transfer parameterisations

Most previously published gas transfer parameterisations are based on $u_{10N}$. In order to compare Eq. (10) with the $u_{10N}$-based parameterisations the measured (EC) $u_*$ was converted to $u_{10N}$ using Eqs. (4), (5), and (6). The linear $u_*$ dependency observed on SOAP corresponds to wind speed dependence that is greater than unity but less than quadratic (Fig. 9). At low to intermediate wind speeds ($u_{10N}$ of $4\,\mathrm{m\,s^{-1}}$–$14\,\mathrm{m\,s^{-1}}$) the SOAP gas transfer coefficients are 0–20% larger than the quadratic Sweeney et al.

5   (2007) parameterisation. Above $14\,\mathrm{m\,s^{-1}}$, however, the SOAP observations are lower than Sweeney et al. (2007), e.g., at $u_{10N} = 20\,\mathrm{m\,s^{-1}}$ the Sweeney et al. (2007) parametrisation predicts 15% higher gas exchange than observed during SOAP. The SOAP gas transfer observations agree well with the COAREG 3.1 bulk flux model at low wind speeds ($u_{10N} < 11\,\mathrm{m\,s^{-1}}$; Fairall et al., 2011). At higher wind speeds, COAREG predicts greater gas exchange than observed during SOAP. At wind speeds ($u_{10N}$) of $13\,\mathrm{m\,s^{-1}}$, $16\,\mathrm{m\,s^{-1}}$, and $20\,\mathrm{m\,s^{-1}}$, COAREG yields 16%, 45%, and 90% higher gas transfer velocities than

10  observed during SOAP, respectively. The recent high latitude Southern Ocean EC measurements from Butterworth and Miller (2016b) agree with the observations from SOAP within the uncertainties of both data sets.





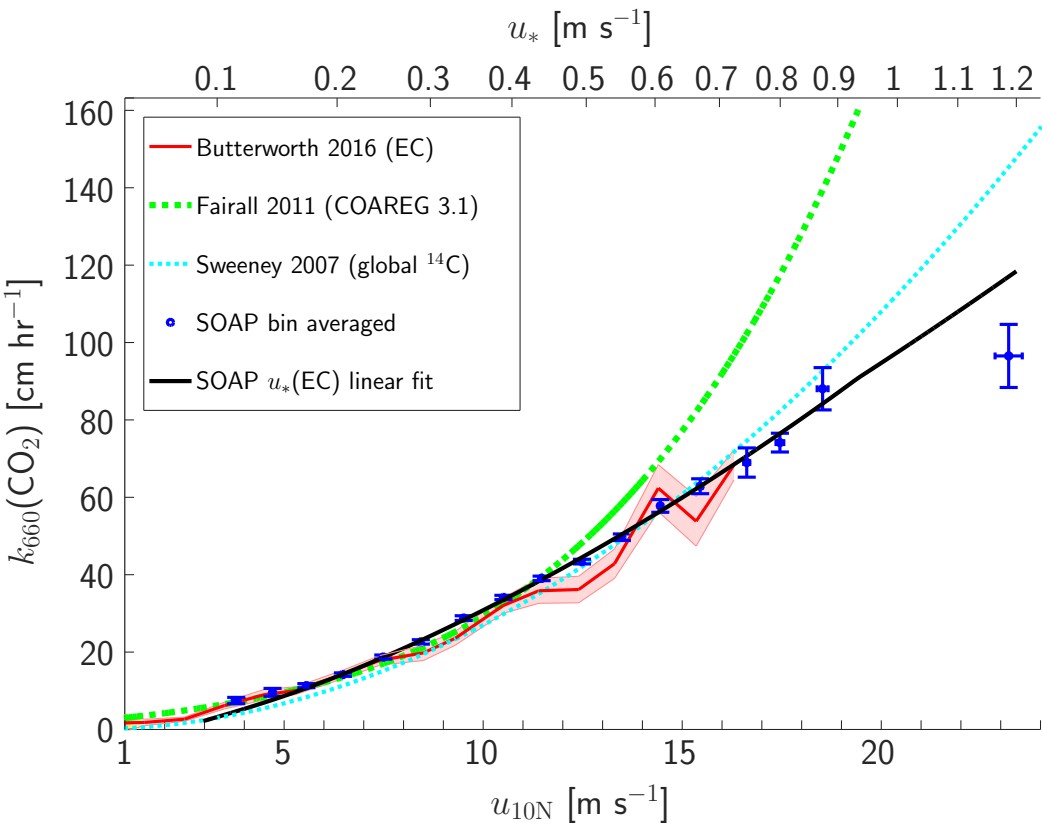

**Figure 9.** $CO_2$ gas transfer velocity normalised to $Sc = 660$ as function of normalised wind speed $u_{10N}$, which was calculated from the directly measured $u_*$ using the COARE 3.5 drag coefficient. The data are bin averaged over $1\,\mathrm{m\,s^{-1}}$ wind speed bins (dark blue) (the width of the last two bins was increased to $3\,\mathrm{m\,s^{-1}}$, in order to account for the scarcity of the data at high wind speeds). The error bars indicate the standard deviations. Equation (10) is shown as black line. Parameterisations from (Sweeney et al., 2007; Fairall et al., 2011) are shown as dashed lines in green, and cyan respectively. Also shown are $1\,\mathrm{m\,s^{-1}}$ wind speed bin median values (and standard deviations) observed by Butterworth and Miller (2016b) with EC in the high latitude Southern Ocean (red).




The SAGE dual tracer ($^3$He/SF$_6$) experiment was conducted in March/April 2004 in the same area as SOAP (Ho et al., 2006, 2007; Smith et al., 2011). In order to compare these data to SOAP, we corrected the QuikSCAT wind speed measurements after Boutin et al. (2009) and converted to $u_*$ using the COARE 3.5 drag coefficient. The transfer velocities were corrected for enhancement of $k$ due to wind speed variability following Wanninkhof et al. (2004), which leads to a 3%-25% reduction of the $k$ values (Smith et al., 2011). The SAGE $k_{600}$ values were converted to $k_{660}$ using (9) with $n = 0.5$. The data from SAGE covers the wind speed range ($u_{10N}$=7–16 m s$^{-1}$). A linear fit to the SAGE data yields,

$$k_{660} = 101.6(\pm16.4)\,u_* - 5.7(\pm7.9), \tag{11}$$

with $k_{660}$ and $u_*$ in the units $[\mathrm{cm\,hr^{-1}}]$ and $[\mathrm{m\,s^{-1}}]$, respectively. The slope and intercept of this relationship are in very good agreement with the SOAP linear fit.

Smith et al. (2011) provided a quadratic fit of the SAGE data to wind speed ($k_{600} = 0.294\,u_{10N}$). Converting this fit to $u_*$(COARE 3.5), yields a goodness of fit similar to that of Eq. (11) ($R^2 = 0.80$ and $R^2 = 0.81$, respectively). However, above $16\,\mathrm{m\,s^{-1}}$, the SAGE quadratic relationship greatly overestimates the SOAP results. (Fig. 10).

Due to the lower $u_{10N}$ estimated by the AFD-corrected wind speed measurements on SOAP (at the bow mast and crowsnest) the usage of those would lead to about 20% and 10% higher $k_{660}$ values at $u_{10N} = 10\,\mathrm{m\,s^{-1}}$ and $u_{10N} = 20\,\mathrm{m\,s^{-1}}$, respectively.




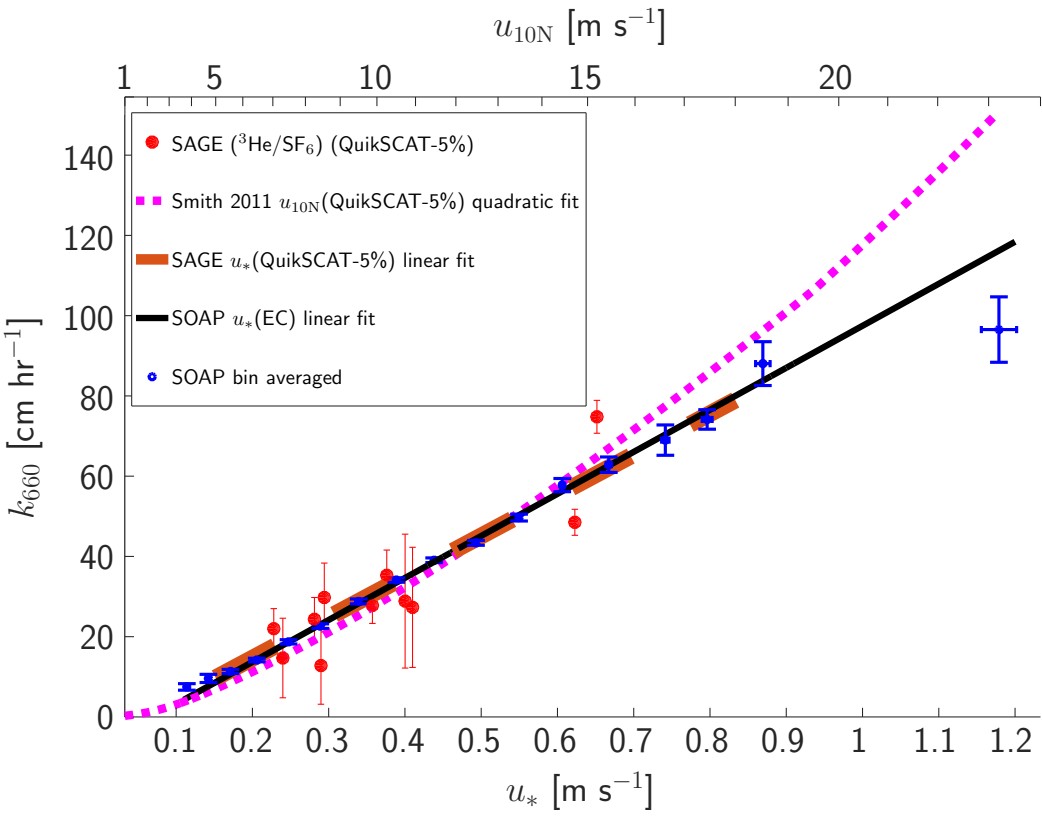

**Figure 10.** Schmidt number normalised gas transfer velocities ($k_{660}$) from the Southern Ocean SOAP and SAGE experiments as a function of friction velocity ($u_*$). Red points - SAGE data from Ho et al. (2007, Table 1), with corrected QuikSCAT wind speeds following Boutin et al. (2009) and converted to $u_*$ using COARE 3.5 (see section 2.5). Magenta dashed line: SAGE parameterisation from Smith et al. (2011). Orange dashed line: $u_*$-linear fit to the SAGE data Eq. (11). Blue points: SOAP shown as $1\,\mathrm{m\,s^{-1}}$ wind speed bin averages. Black line: SOAP $u_*$-linear regression (Eq. 10). The error bars indicate the standard error of the mean.





## 5  Conclusions

Direct eddy covariance $CO_2$ and momentum flux measurements made on board the R/V-*Tangaroa* during the SOAP experiment have been reanalysed using a series of established and new corrections for platform motion, air-flow distortion, and sensor cross-sensitivity. Reprocessing the SOAP data resulted in $CO_2$ gas transfer velocities with considerably less scatter than prior studies using similar instrumentation. The improved SOAP data set exhibits a strong linear correlation between $CO_2$ transfer

5    velocity and friction velocity over a wind speed ($u_{10N}$) range of $5 - 18\,\mathrm{m\,s^{-1}}$. This result is surprising, and suggests that the contribution of bubble-mediated $CO_2$ gas transfer may be overestimated in current physically-based gas transfer models, or that a reduction of the interfacial gas transfer at high wind speeds may offset the bubble-mediated enhancement (e.g. Soloviev, 2007).

A method was presented to estimate the uplift of air over the ships bow from the measured cospectra. The uplift estimate is

10   used to improve the adjustment of bulk measurements, like wind speed or temperature, to a standard height (10 m a.s.l.).

Further improvement in the analysis of shipboard eddy covariance data is possible with improved motion and ship navigational corrections and a physically-based understanding of a high frequency bias in the tilt-corrected momentum flux cospectra.



## Appendix A: Vertical tilt and uplift estimation

When the air stream approaches the ship the stream lines are distorted by the bluff body. This leads to an uplift of the air from its original height and to an upward tilt of the streamlines. For accurate Eddy Covariance flux estimates it is essential to estimate to tilt of the wind vector. Figure A1 shows the vertical tilt estimates that were used in Landwehr et al. (2014) and the estimates obtained following Landwehr et al. (2015), as a function of the relative wind direction. For underway data the incorrect order

5    of motion and tilt-correction used in Landwehr et al. (2014) lead to large overestimations of the tilt and consequently biased EC flux results. The LES model predictions for the vertical tilt (Popinet et al., 2004) are of the same magnitude but show a contrary functionality with relative wind direction for $|\alpha| \leq 60°$. Note that the bow mast was not included in the LES model of the R/V-*Tangaroa*. This might explain the differences between the observed tilts and the LES simulation.





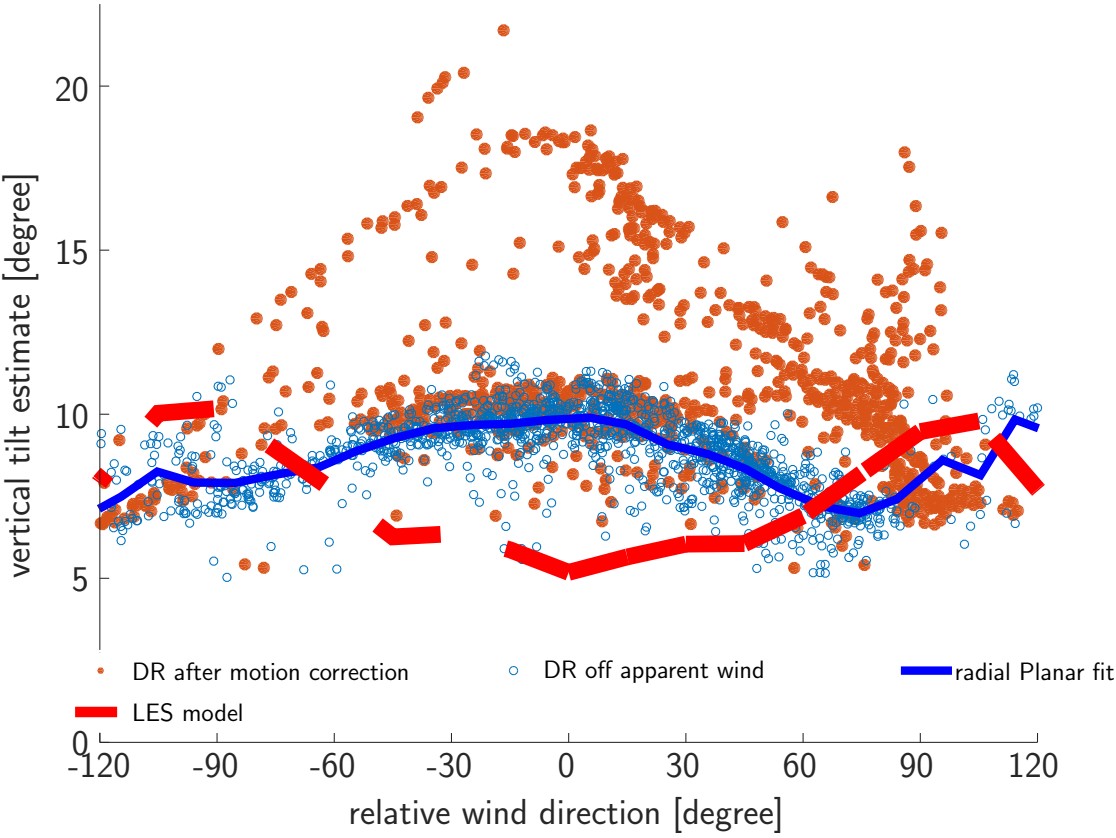

**Figure A1.** Estimates of the vertical tilt of the wind vector from the starboard side anemometer as function of the relative wind direction. (i) using the true wind speed as done in Landwehr et al. (2014) (orange dots); (ii) using the apparent wind speed as suggested by Landwehr et al. (2015) (blue circles); (iii) using the radial planar fit (Landwehr et al., 2015) (blue line); (iv) from the LES model of the R/V-*Tangaroa* (Popinet et al., 2004) (red dashed line).





Due to the uplift of the air flow passing over the ship, the true measurement height $\tilde{z}$ does not coincide with the average height $z$ of the anemometer on board. While not affecting the measurement of air-sea fluxes with the EC method, the true measurement height and thus the uplift $\Delta z = z - \tilde{z}$ is an important parameter for the normalisation of the wind speed $u_{10\mathrm{N}}$ and for the interpretation of the observed cospectra.

Here the observed cospectra were used to estimate the effective measurement height and thus the uplift. The momentum flux
spectra were averaged over 1–2 hour intervals with steady speed and heading of the ship. A total of 95 such intervals where found. The $\mathrm{nCo}_{\langle uw \rangle}(n)$ were fitted with the universal shape of the momentum flux cospectrum, proposed by Kaimal et al. (1972):

$$\frac{nCo_{uw}(n)}{\langle uw \rangle} = \frac{A\frac{f}{f_0}}{1 + B(\frac{f}{f_0})^C} \tag{A1}$$

where $A = 0.88$, $B = 1.5$, the exponent $C = 2.1$, and the characteristic non-dimensional frequency is given by $f_0 = 0.1\left[G(\frac{z}{L})\right]^{3/4}$,
were $G$ is a function of the non-dimensional stability parameter $\zeta = \frac{z}{L}$. The square of the residuals (weighted with the standard deviation of the frequency weighted cospectral averages) were minimised by varying $A$ and $n_0 = f_0\frac{U}{z}$, while keeping $B = 1.5$ and $C = 2.1$ constant. The average of the fit was found to be $A = 0.86 \pm 0.07$, which agrees within uncertainty with the value of $A = 0.88$ found by Kaimal et al. (1972). For all intervals with unstable to neutral stability ($\frac{z}{L} \leq 0$ and $G(\frac{z}{L}) = 1$) the original height of the measured streamlines can be estimated with,

$$\tilde{z} = 0.1\frac{z_{\mathrm{bow}}}{f_0} = 0.1\frac{U}{n_0}, \tag{A2}$$

where $z_{\mathrm{bow}} = 12.6\,\mathrm{m}$ is the nominal measurement height above mean sea level. The estimates of $\tilde{z}$ were bin-average over $15°$ absolute wind direction bins (assuming symmetry over the ships main axis). The results are plotted in Fig. A2 and compared with the uplift estimates from the LES model (Popinet et al., 2004).

For wind direction at $0°$ to the bow, the resulting $\tilde{z}$ agreed (within the uncertainties) with the predictions of the LES sim-
ulation (Popinet et al., 2004). For $20° \leqslant |\alpha| \leqslant 60°$ the fit to cospectra indicates a higher uplift than the LES model, while for $|\alpha| \geqslant 60°$ the observed uplift is lower than predicted by the LES model.




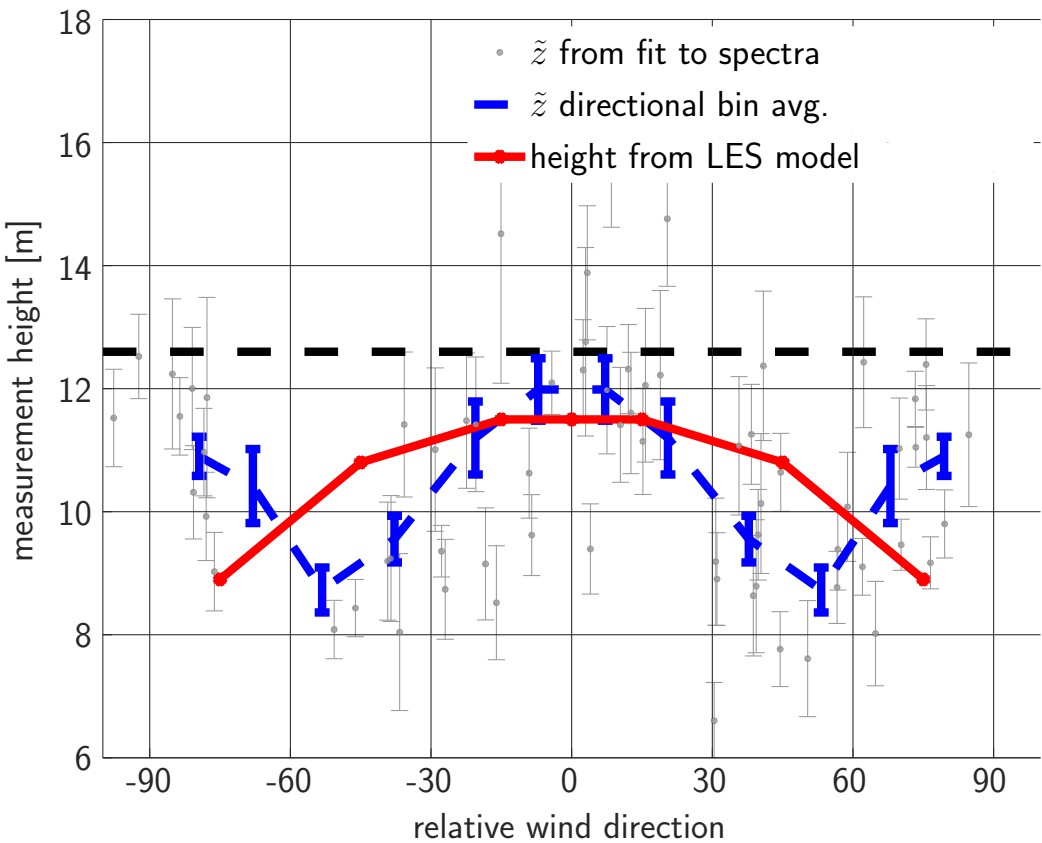

**Figure A2.** Measurement height estimated with (A2) as function of the relative wind direction. Individual measurements are shown as dots with the errorbars indicating the uncertainty of the fit. The measurements are averaged over direction bins, this is shown as dashed blue line. The dashed black line indicates the nominal measurement height $z_{\mathrm{bow}} = 12.6\,\mathrm{m}$ and the red curve shows the results from the Gerris Large Eddy Simulation. Only results obtained from fits to unstable spectra $\frac{z}{L} \leq 0$ (with $G(\frac{z}{L}) = 1$) are shown in this plot and were used to estimate $\tilde{z}$.





## Appendix B: Shape of the cospectra

The size distribution $nCo_{uw}(\lambda)$ of the turbulent energy depends on the stability parameter $\zeta$ (Kaimal et al., 1972). In an EC system, however, the turbulences are recorded as time series. The frequency distribution $nCo_{uw}(f)$ reflects the size distribution depending on the relative velocity ($f \sim \lambda U$). For a stationary observer $U = |\mathbf{u}|$, the true wind speed; but for a moving observer it is necessary to take the observer's velocity ($\mathbf{v}_{\text{obs.}}$) into account ($U = |\mathbf{u} + \mathbf{v}_{\text{obs.}}|$). This is illustrated in Fig. B1, where average normalised cospectra are shown for three scenarios:

a) with $u_a = 9.3\,\text{m}\,\text{s}^{-1}$, $v_{\text{obs.,a}} = 0\,\text{m}\,\text{s}^{-1} \longrightarrow U_a \approx 9.3\,\text{m}\,\text{s}^{-1}$,

b) with $u_b = 9.3\,\text{m}\,\text{s}^{-1}$, $v_{\text{obs.,b}} = 4\,\text{m}\,\text{s}^{-1}$, $\longrightarrow U_b \approx 13.5\,\text{m}\,\text{s}^{-1}$,

c) with $u_c = 13.0\,\text{m}\,\text{s}^{-1}$, $v_{\text{obs.,c}} = 0\,\text{m}\,\text{s}^{-1}$, $\longrightarrow U_c \approx 13.5\,\text{m}\,\text{s}^{-1}$.

This provides $u_a \approx u_b \approx 9.3\,\text{m}\,\text{s}^{-1}$ and $U_b \approx U_c \approx 13.5\,\text{m}\,\text{s}^{-1}$. The average cospectra from case (b) and (c) are similar and shifted to higher frequencies when compared to case (a). This shows that the relative wind speed $U$, rather than the true wind speed $u$, determines the frequency distribution of the turbulent energy.

Figure B2 shows the normalised cospectra of the momentum $CO_2$ and sensible heat flux grouped for atmospheric stability as function of the non-dimensional frequency (using $U$, $L$, and the directional dependent estimates of $\tilde{z}$). As described in section 2.5, for $f > 1$ the energy observed in $nCo_{uw}$ is higher than expected from the universal shape. The estimated effect on $u_*$ was however relatively small (0-6%, on average 2% overestimation, see Fig. B3). However in general the cospectra exhibit Kaimal like shapes, mostly follow -4/3 or slope, and shift to higher frequencies for $\zeta > 0$ as expected. Note that the shown spectra are only weighted with the corresponding EC flux and do therefore not collapse in the inertial sub-range.



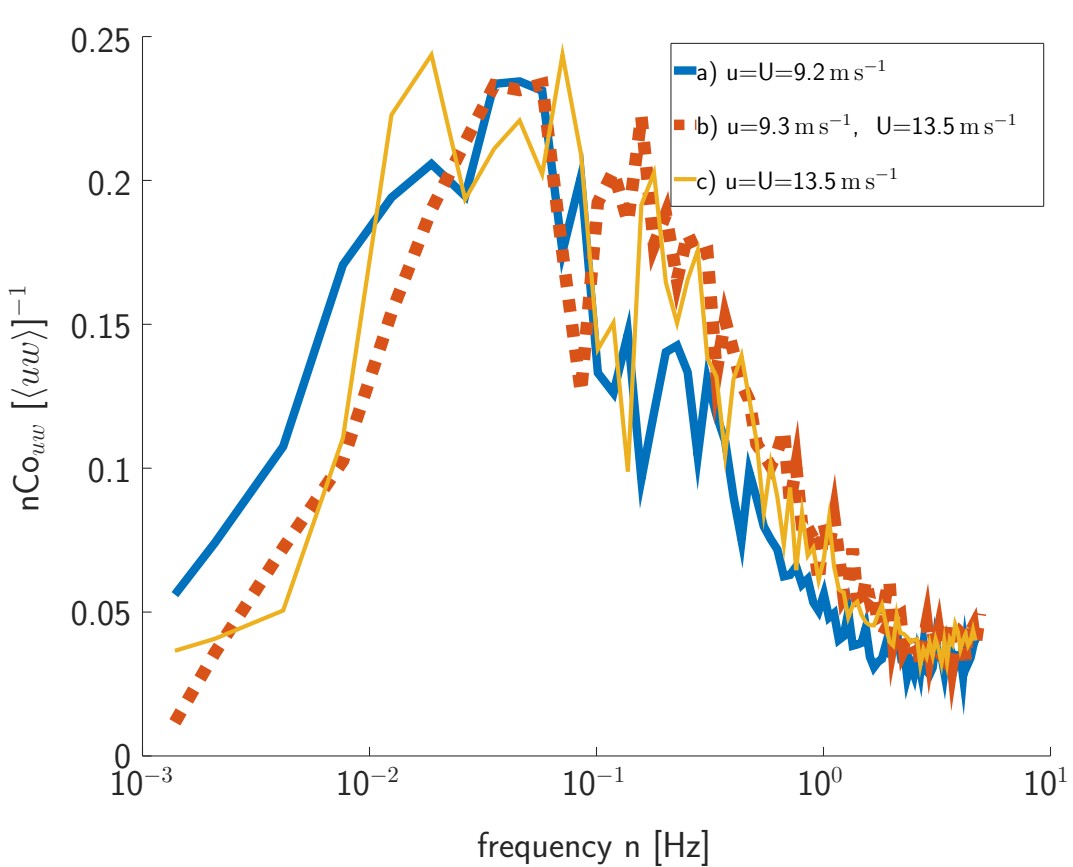

**Figure B1.** Average normalised cospectra of the momentum flux (a) station measurements with $8\,\mathrm{m\,s^{-1}} < u_{10\mathrm{N}} < 10\,\mathrm{m\,s^{-1}}$, (b) underway measurements with $8\,\mathrm{m\,s^{-1}} < u_{10\mathrm{N}} < 10\,\mathrm{m\,s^{-1}}$ and a ship's speed of $v_{obs.} \approx 4\,\mathrm{m\,s^{-1}}$, (c) station measurements with $12\,\mathrm{m\,s^{-1}} < u_{10\mathrm{N}} < 14\,\mathrm{m\,s^{-1}}$. Only data with relative wind direction $|\alpha| < 20°$ and $z/L < 0$ are used.





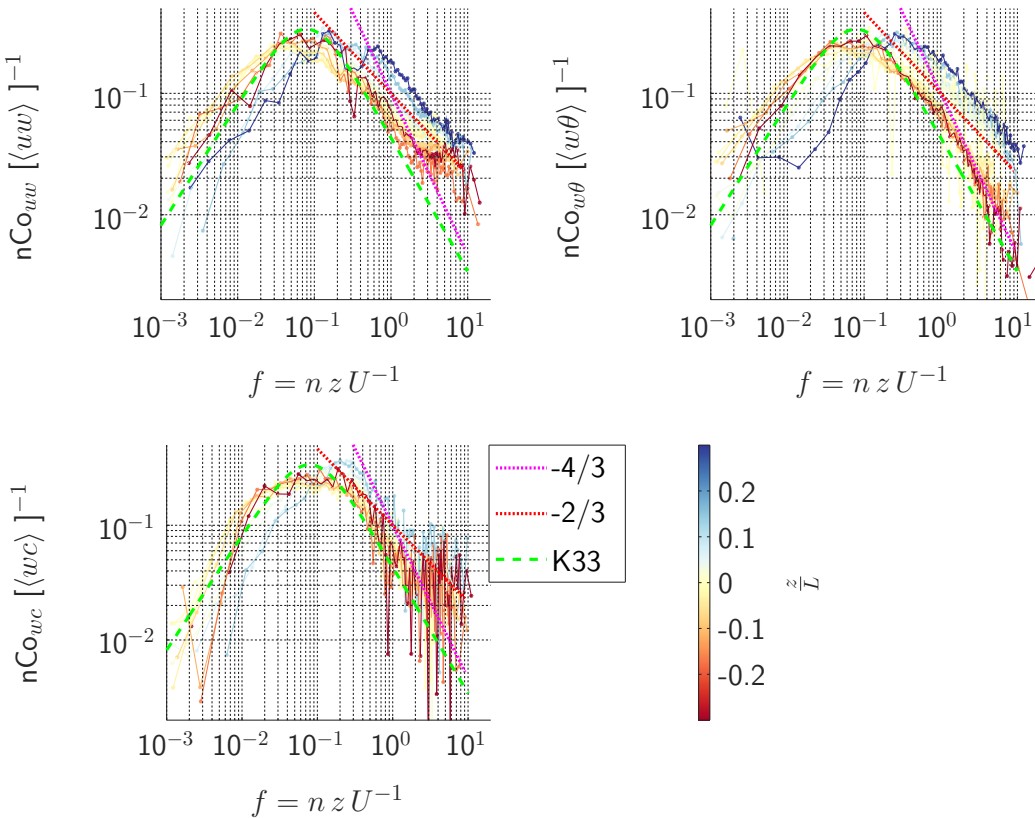

**Figure B2.** Normalised Cospectra of momentum $CO_2$ and sensible heat bin averaged over based on the dimensionless stability parameter. The dashed green curve shows (A1) (Kaimal et al., 1972). The magenta and red dotted lines indicate the expected slopes in the inertial sub-range for Co and power spectra respectively. Note that the cospectra are not normalised for the stability function, this would reduce the magnitude of the stable spectra and make them fall together with the unstable-neutral spectra in the inertial subrange.



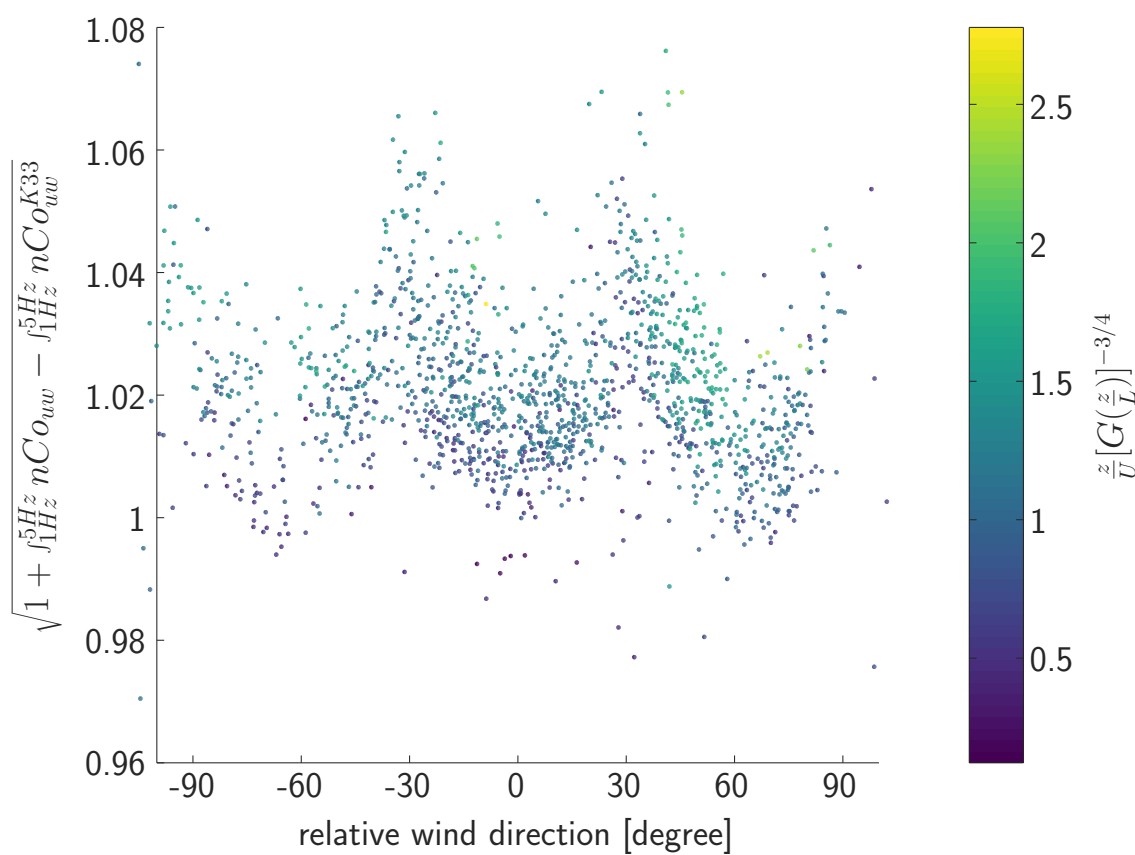

**Figure B3.** Overestimation of $u_*$ due to elevated cospectral density observed at $n \geqslant 1\,\mathrm{Hz}$ estimated by comparing $\mathrm{nCo}_{uw}(n \geq 1\,\mathrm{Hz})$ with (A1) as function of the relative wind direction and the value of the non-dimensional frequency $f_{n=1\,\mathrm{Hz}}$ corresponding to $n = 1\,\mathrm{Hz}$.





## Appendix C: Flux calculations and Data quality assessment

The data were separated in 12 minute long intervals, over which all averages and fluxes were computed. For the correlation of $CO_2$ fluctuations measured in the closed-path analyser with $w'$ the optimal time lag was found by searching for the maximum covariance $\langle w'(t)x'_{CO_2}(t+\delta t)\rangle$ within a reasonable range of $\delta t = 0 - 3\,\mathrm{seconds}$. For both IRGAs *dryA* and *dryB* the average optimal time lag was $\langle\delta t\rangle = 1.50(\pm 0.15)\,\mathrm{seconds}$. Individual $\delta t$ that deviated more than two samples from $\langle\delta t\rangle$ were defaulted

to $\delta t = \langle\delta t\rangle$. This was the case only for periods of relatively low $CO_2$ fluxes and during the high wind speed event doy=(60.76-60.80), where residual motion related signals lead to a strong correlation between $w$ and $x_{CO_2}$ at $\delta t = 0$.

Power and cospectra are computed as fast Fourier transforms (FFT). All spectra are smoothed and down sampled using linearly increasing averaging intervals in the frequency domain.

The quality control of the data were performed in different levels. For *stage A* momentum flux data were rejected when any

of the following criteria was fulfilled:

– The vector average of the instantaneous course/heading vector was smaller than 0.90 (1 indicates a perfectly stable course). This corresponds to a maximum standard deviation of the heading of 25°.

– The measured relative wind direction to the bow $|\langle\alpha\rangle| \geq 110°$

– The true relative wind direction to the bow $|\langle\alpha_{\mathrm{true}}\rangle| \geq 125°$

– Measured average relative wind speed $\langle u_{\mathrm{me}}\rangle \leq 1\,\mathrm{m\,s^{-1}}$

– Data from 20-Feb-2012 18:30 till 21-Feb-2012 06:00 were excluded for malfunctioning of the starboard anemometer.

– If the momentum flux computed over the 12 minutes period was more than 30% different from the average momentum flux computed over 5 subintervals of 2.4 minutes.

This removed approx 30% of the total number of 2222 available measurements. Only a fraction of 2% were excluded solemnly

based on the stability test.

The $CO_2$ flux data were discriminated based on the following *stage A* criteria:

– If the momentum flux failed the stage A quality control.

– The air-sea $CO_2$ concentration difference was low ($|\Delta pCO_2| \leq 30\mathrm{ppm}$).

– The root mean square value (RMS) of $x_{CO_2}$ was larger than 0.3 ppm (the total median and restricted median and mean

values of the RMS were 0.07 ppm and 0.06 ppm, for LI-7500 *dryA* and *dryB*, respectively).

– Strong stable atmospheric condition ($\frac{z}{L} \geq +0.2$) where excluded to reduce the uncertainty introduced by the high frequency loss correction (see section 2.7).





This removed 45% of the total number of available measurements. Further a total of 8 measurements with negative transfer velocities were excluded.

The quality control *stage B* was mainly based on the shape of the cumulative sum of normalised cospectra ($F_{sum}$) as function of the non-dimensional frequency with stability correction $f = n\frac{z}{U}[G(\frac{z}{L})]^{-3/4}$, where $G$ is taken from (Kaimal et al., 1972), to account for the shift of the spectra to higher frequencies for $\frac{z}{L} >> 0$. Intervals where any of the following criteria was fulfilled for the normalised along-wind momentum flux cospectrum, where excluded.

- $F_{sum} \leq -0.2$ @ $f = 0.03\,\mathrm{Hz}$

- $F_{sum} \geq +0.7$ @ $f = 0.03\,\mathrm{Hz}$

- $F_{sum} \geq +0.9$ @ $f = 0.1\,\mathrm{Hz}$

- $F_{sum} \leq +0.8$ @ $f = 1\,\mathrm{Hz}$

- $\min(F_{sum}) \leq -0.2$

- $\max(F_{sum}) \geq 1.1$

- $\min(nCo^{norm}) \leq -1$

- $\max(nCo^{norm}) \geq 2$

The same filter was applied to the heat flux cospectra which where used for the estimation of the high frequency flux loss in section 2.7.

Accounting for the lower signal to noise ration in the $CO_2$ flux spectra and the effects of high frequency attenuation, only the last four of the above criteria where applied as additional filter on the $CO_2$. For both the momentum and $CO_2$ flux, stage B removed about 6% of the data that had passed the respective stage A.

Using $\tilde{z}$, $U$, and $L$ with (A1) allows to estimated how much of the turbulent flux signal would be expected to be outside of the observed frequency range $1/720$ to $5\,\mathrm{Hz}$. Based on (A1), 98.3% of the theoretical momentum flux spectrum was resolved for $U \approx 15\,\mathrm{m\,s^{-1}}$ and $\frac{z}{L} \geq 0$. Between 98% and 98.3% of $nCo_{uw}^{K33}$ were resolved for $15\,\mathrm{m\,s^{-1}} \leq U \leq 25\,\mathrm{m\,s^{-1}}$. Thus the theoretical loss cause by the limited measurement frequency of $10\,\mathrm{Hz}$ was always less than 2%. Even for low wind speeds ($U \leq 3\,\mathrm{m\,s^{-1}}$) and unstable stratification, the theoretically resolved fraction was not lower than 96%. The chosen flux-averaging time of 12 minutes was therefore adequate to resolve the turbulent air-sea fluxes.



## 25    Appendix D:  Differences in wind speed and momentum flux estimates from the two bow mast anemometer

The relative difference between the momentum flux/wind speed measurements of the port and starboard anemometer illustrates the small scale variability of the flow-distortion effects, but also gives an indication on the absolute flow-distortion errors in the measurement of each anemometer. Figure D1 shows the relative difference of the port and starboard anemometer measurements of friction velocity and wind speed as function of the relative wind direction. The wind speed as well as the friction velocity

5   estimates from the two anemometer agree well with each for bow on relative wind directions. However for increasing relative wind direction the windward anemometer reads up to 7% and 6% higher wind speed $U$ and friction velocity $u_*$ than the leeward anemometer, respectively. The air flow distortion correction with the model results from (Popinet et al., 2004) removes only 30% of the relative difference in the wind speed measurements from the two bow mast anemometers. The MSC and NAV regression corrections clearly reduce relative differences observe in the $u_*$ measurement from the two anemometers to within

10   $\pm 2\%$ for most wind direction sectors and most of the measurements. With all corrections applied the relative difference in the $u_*$ estimates from the two anemometers is less than half of the relative difference of the wind speed estimates.



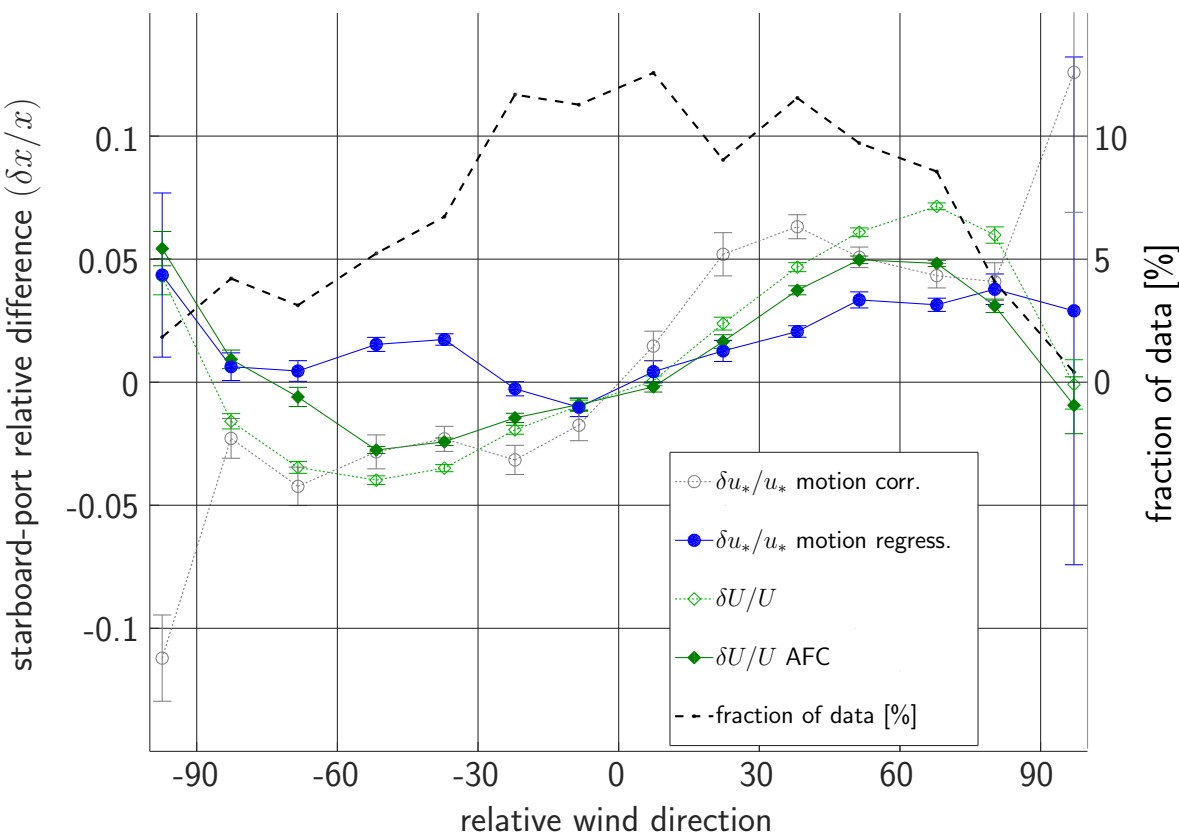

**Figure D1.** Relative differences of starboard and port side measurements of $u_*$ and wind speed as a function of the relative wind direction calculated as (stbd-port)/(stbd+port) and averaged over $15°$ wind direction bins. The plot shows $\delta u_*/\langle u_* \rangle$ for only motion and tilt corrected wind speeds (grey dashed open circles) and with all regression corrections applied (blue filled circles), respectively. The relative difference of the wind speed measurements, with and without air flow distortion correction applied, are shown as open and filled green diamonds, respectively. The black dashed line indicates the fraction of data observed in each wind direction sector. The errorbars show the standard error of the mean values.





*Acknowledgements.* We thank Cliff Law (NIWA), for his leadership of the SOAP field campaign, Kim Currie (NIWA) who provided the underway $\Delta pCO_2$ data, Cyril McCormic for technical assistance, and the captain and crew of the R/V *Tangaroa* for their support in the field. The SOAP shipboard field program and underway $CO_2$ measurements were funded in New Zealand under NIWA's Atmosphere Research Programme 2. The seagoing $CO_2$ flux measurements are a contribution to US SOLAS and were supported by the U.S. NSF Atmospheric Chemistry (grants 0851407, 0851472 and 1143709) and IR/D programs. The Science Foundation Ireland supported this work as part of the US-Ireland R&D Partnership Programme (grant 08/US/I1455) and through a Short-term Travel Fellowship (grant 09/US/I1758- STTF-11). The Norwegian Research Council provided support under the projects 233901 and 244262. Tom Bell received additional support from the UK NERC (grant NE/N018095/1).



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
