# Peer review of "Using Eddy Covariance to Measure the Dependence of Air-Sea CO2 Exchange Rate on Friction Velocity"

_Atmospheric Chemistry and Physics, 2017_

## Referee Comment (RC1) · Anonymous Referee #1 · 26 Nov 2017

The paper is well written with a clear structure. The methodology is well described and the results are interesting. To summarize; the paper is in good shape and will make an interesting contribution to the Atmospheric Chemistry and Physics Discussions.

In order to further improve the paper, e.g., the following could be elaborated:

Section 2.2. and related sections Ship motions are generally described by the six degrees of freedom that a ship can experience, while the ship in the large eddy simulation (Popinet et al., 2004) is stationary. The large eddy simulation uses numerical dissipation instead of a sub-grid scale turbulence model. An uniform inflow velocity is applied at the inlet boundary. These three aspects influence the results of the large eddy sim-

ulation and therefore in turn the calculation of the uplift. Please elaborate how these aspects are accounted for in the correlations?

Section 2.5. Page 10 (9-13): It is here suggested that it may be that small-scale turbulence adjusts to the new orientation of the tilted stream lines. Small-scale turbulence is typically more iso-tropic than large-scale turbulence. It is therefore counter intuitive that it rather is the small-scale than the large-scale turbulence that is adjusted to the tilted stream lines. It may, however, be that the magnitude of the small-scale turbulence is increased in areas of increased shear as a result of a tilted air flow. It may also be that since the small-scale turbulence is more iso-tropic than large-scale, it is not so influenced by the tilted air flow. Please clarify what is meant by the word adjust in this discussion. Page 11 (9-22) This description does not belong to this section. Please consider moving it to another section or make it an own section.

Section 2.8. Page 16 (12-13): The transition of n is discussed in terms of smooth and rough due to increasing wind speed. n is also to a large degree dependent on surfactants, especially during low wind conditions, e.g., [Frew et al., 2004; McKenna and McGillis, 2004; Zhang et al., 2013]. Please add some references regarding this phenomenon.

Section 3.3 Page 23 (2) wind stress definition should be with squared friction velocity.

Section 4.1. Page 30 (15-16) This section discusses how the gas transfer velocity relates to the friction velocity and that buoyancy-driven processes may contribute significantly at lower wind speeds. E.g., [Fredriksson et al., 2016] discuss the transition between a gas transfer velocity mainly driven by buoyancy processes to a gas transfer velocity mainly driven by shear stress processes via the Richardson number (relates the buoyancy flux to the friction velocity). This paper can be used in the discussion regarding the range of friction velocity, where the gas transfer velocity as a function of friction velocity is valid.

Fredriksson, S. T., L. Arneborg, H. Nilsson, and R. A. Handler (2016), Surface shear

stress dependence of gas transfer velocity parameterizations using DNS, Journal of Geophysical Research: Oceans, n/a-n/a,10.1002/2016JC011852. Frew, N. M., et al. (2004), Air-sea gas transfer: Its dependence on wind stress, small-scale roughness, and surface films, J Geophys Res-Oceans, 109(C8),Artn C08s17,Doi 10.1029/2003jc002131. McKenna, S. P., and W. R. McGillis (2004), The role of free-surface turbulence and surfactants in air-water gas transfer, Int J Heat Mass Tran, 47(3), 539-553,DOI 10.1016/j.ijheatmasstransfer.2003.06.001. Zhang, Q., R. A. Handler, and S. T. Fredriksson (2013), Direct numerical simulation of turbulent free convection in the presence of a surfactant, Int J Heat Mass Tran, 61, 82-93,DOI 10.1016/j.ijheatmasstransfer.2013.01.031.

---

## Referee Comment (RC2) · Anonymous Referee #2 · 28 Nov 2017

This paper presents a new corrections of the estimated air-sea CO2 gas flux in ship measurements considering the air-flow distortion. And the air-sea CO2 gas transfer velocity using this corrections showed the smaller variation than the former result. This manuscript contains possibly interesting points for the readers of Atmospheric Chemistry and Physics. However, the Result and Discussion part is not described adequately. The following are the concerns and some suggestions.

In Section 3.2, the authors mentioned the residual flow distortion error about the disagreement with COARE 3.5 parameterization. Consequently, the authors should describe the reasons in detail.

[Figure]

In Section 3.3, The authors should clearly mention about the accuracy of catamaran's data. For example, what is the method of the motion correction for the catamaran's data? And how about the accuracy of the catamaran's wind speed?

In Section 4.1, since the $CO_2$ gas transfer velocity decrease in the high wind speed in Fig. 8, it is better to add the applicable wind speed range to Eq. (10).

---

## Referee Comment (RC3) · Anonymous Referee #3 · 29 Nov 2017

This paper is a careful reanalysis of the eddy covariance measurements of momentum and carbon dioxide fluxes from the Southern Ocean Surface Ocean Aerosol Production study (SOAP). Every aspect of motion correction and flow distortion on the means and fluxes is considered and appropriate corrections applied. The smooth dependence on wind speed, of the friction and mass transfer velocities, suggests that the corrections are accurate and complete. The slightly higher friction velocities than those from COARE 3.5 may be a fetch effect: COARE was open ocean while SOAP was shorter fetch corresponding to higher drag coefficients. The divergence from quadratic of the CO2 mass transfer velocity above wind speed of 16 m/s has important repercussions for the global carbon budget and is the main scientific (rather than technical) product

of this fine work.

---

## Author Comment (AC1) · 29 Dec 2017

**Response to Referee $\#1$**

**RC** : *The paper is well written with a clear structure. The methodology is well described and the results are interesting. To summarize; the paper is in good shape and will make an interesting contribution to the Atmospheric Chemistry and Physics Discussions. In order to further improve the paper, e.g., the following could be elaborated:*

**AC** : We would like to thank the reviewer for this encouraging feedback and the constructive comments, to which we will respond in detail below.

[Figure]

**RC** : *Section 2.2. and related sections Ship motions are generally described by the six degrees of freedom that a ship can experience, while the ship in the large eddy simulation (Popinet et al., 2004) is stationary. The large eddy simulation uses numerical dissipation instead of a sub-grid scale turbulence model. An uniform inflow velocity is applied at the inlet boundary. These three aspects influence the results of the large eddy simulation and therefore in turn the calculation of the uplift. Please elaborate how these aspects are accounted for in the correlations?*

**AC** : The reviewer correctly names the main shortcomings of the LES. We have acknowledged these shortcomings in Section 2.2. The deviations of the LES results from the observations may be attributed to these simplifications, as well as unresolved structures (such as mast supports) in the model. It was, however, not the focus of this work to improve the LES model and we did not investigate quantifying the uncertainties arising from the simplifications made. Here we present two additional methods, which use 3D wind speed measurements to validate air-flow models, aside from the difference of mean velocity and mean horizontal deflection measured at two different points. These are (i) the vertical tilt of the wind vector, which is measured from the ratio of mean vertical and horizontal velocities, and (ii) the vertical uplift, which can be estimated from the frequency distribution of the observed turbulences.

**RC** : *Section 2.5. Page 10 (9-13): It is here suggested that it may be that small-scale turbulence adjusts to the new orientation of the tilted stream lines. Small-scale turbulence is typically more iso-tropic than large-scale turbulence. It is therefore counter intuitive that it rather is the small-scale than the large-scale turbulence that is adjusted to the tilted stream lines. It may, however, be that the magnitude of the small-scale turbulence is increased in areas of increased shear as a result of a tilted air flow. It may also be that since the small-scale turbulence is more iso-tropic than large-scale, it is not so influenced by the tilted air flow. Please clarify what is meant by the word adjust in this discussion.*

**AC** : The sentence is indeed misleading (turned out wrong) we apologise and suggest rewording as follows: "Our observation in described in Sec. 2.5 suggests that while rotation of the coordinate system into the air stream is crucial to adequately measure the contribution of large eddies, it is counter-productive for the measurement of flux carried by small eddies, i.e., it may be that the small scale turbulence ($\lambda < 1\,\mathrm{m}$) does not adjust its orientation to the new flow direction as efficiently as the large scale turbulence. Another possible explanation could be that the magnitude of small scale turbulence may be increased locally as a result of the shear in the tilted and accelerated air flow." We did also expanded on the following "Due to the projection of the auto-covariances of the three components $u, v$, and $w$, the momentum flux estimate is generally more sensitive to the choice of the coordinate system than the scalar fluxes (see Wilczak et al., 2001). The sensitivity of the $\mathrm{nCo}_{uw}(n)$ estimate to the tilt increases in the inertial sub-range, where the auto-covariances of the three velocity components diminish slower with increasing frequency ($f^{-2/3}$)."

**RC** : *Section 2.5. Page 11 (9-22) This description does not belong to this section. Please consider moving it to another section or make it an own section.*

**AC** : We moved this to a new Section 3.2.

**RC** : *Section 2.8. Page 16 (12-13): The transition of n is discussed in terms of smooth and rough due to increasing wind speed. n is also to a large degree dependent on surfactants, especially during low wind conditions, e.g., [Frew et al., 2004; McKenna and McGillis, 2004; Zhang et al., 2013]. Please add some references regarding this phenomenon.*

**AC** : We have added the following sentence to acknowledged this phenomenon:
"The exact shape wind speed dependence of this transition has been found in to depend on surfactant concentration on the water surface (e.g. Frew et al., 2004; McKenna and McGillis, 2004; Zhang et al., 2013; Krall, 2013)."

**RC** : *Section 3.3(3.4) Page 23 (2) wind stress definition should be with squared friction velocity.*

**AC** : Thanks for spotting this mistake, it is now corrected.
**RC** : *Section 4.1. Page 30 (15-16) This section discusses how the gas transfer velocity relates to the friction velocity and that buoyancy-driven processes may contribute significantly at lower wind speeds. E.g., [Fredriksson et al., 2016] discuss the transition between a gas transfer velocity mainly driven by buoyancy processes to a gas transfer velocity mainly driven by shear stress processes via the Richardson number (relates the buoyancy flux to the friction velocity). This paper can be used in the discussion regarding the range of friction velocity, where the gas transfer velocity as a function of friction velocity is valid.*

**AC** : The results of Fredriksson et al. (2016) suggest that the influence of the buoyancy flux on air-water gas exchange may become relevant at low wind speeds, depending on the magnitude of the surface buoyancy flux. For wind speeds below $5\,\mathrm{m\,s^{-1}}$, the magnitude of the surface heat flux remained below $200\,\mathrm{W\,m^{-2}}$ during SOAP. Our estimates of the Richardson number ($Ri = B_{0,w}\nu_w u_{*,w}^{-4}$) were below the critical value of $Ri \approx 0.004$, suggested by Fredriksson et al. (2016). However, we would like to note here, that the slope of Eq. (10) is more than a factor of two higher than typical parametrisations of shear driven gas transfer (e.g. Jähne et al., 1987) and that Eq. (10) predicts negative gas exchange rates at low wind speeds. It is therefore likely that the observed $CO_2$ transfer velocities are from the combined effect of shear driven gas transfer as well as of wave breaking and bubble-driven gas transfer. It is however not within the scope of this paper to present a physical based equation, appropriately describes these contributions.

We have modified the part of Sec. 4.1 as follows:

"Eq. (10) should not be interpreted as physical law, but rather as empirical parametrisation for the wind speed range $5 - 19\,\mathrm{m\,s^{-1}}$. Extrapolation of this linear $k$ vs EC $u_*$ relationship outside of the wind speed range of the SOAP data set is not recommended, because there are physical reasons why this relationship might not hold. At lower wind speeds, buoyancy-driven processes may contribute significantly to gas transfer (Soloviev, 2007; Fredriksson et al., 2016). In fact (10) slightly underestimates the wind speed binned data for $u_{10N} < 5\,\mathrm{m\,s^{-1}}$) and would predict negative $k_{660}$ for $u_* \leq 0.07\,\mathrm{m\,s^{-1}}$ ($u_{10N} \leq 2.3\,\mathrm{m\,s^{-1}}$)). However, since our estimations of the Richardson number ($Ri = B_{0,w}\nu_w u_{*,w}^{-4}$) remained below the critical value of $Ri \approx 0.004$,

which was suggested by Fredriksson et al. (2016), we do not expect significant contribution of buoyancy-driven processes to the gas exchange rates observed during SOAP. Here $B_{0,w}$, $\nu_w$, and $u_{*,w}$ are the water side surface buoyancy flux, kinematic viscosity of sea water, and waterside friction velocity respectively. At higher wind speeds, wave breaking and bubble-driven gas transfer are expected to contribute to gas transfer of $CO_2$ and other sparingly soluble gases (Woolf, 1997; Fairall et al., 2011; Bell et al., 2017). Surprisingly, there is no evidence in the SOAP data for an increase in the slope of the $k_{660}$ vs $u_*$ relationship at high wind speeds. If anything, the limited SOAP data available at the highest wind speeds appear to be biased low relative to the linear regression."

**References**

Bell, T. G., Landwehr, S., Miller, S. D., de Bruyn, W. J., Callaghan, A. H., Scanlon, B., Ward, B., Yang, M., and Saltzman, E. S.: Estimation of bubble-mediated air–sea gas exchange from concurrent DMS and $CO_2$ transfer velocities at intermediate–high wind speeds, Atmospheric Chemistry and Physics, 17, 9019–9033, doi:10.5194/acp-17-9019-2017, https://www.atmos-chem-phys.net/17/9019/2017/, 2017.

Esters, L., Landwehr, S., Sutherland, G., Bell, T. G., Christensen, K. H., Saltzman, E. S., Miller, S. D., and Ward, B.: Parameterizing air-sea gas transfer velocity with dissipation, Journal of Geophysical Research: Oceans, pp. n/a–n/a, doi:10.1002/2016JC012088, http://dx.doi.org/10.1002/2016JC012088, 2017.

Fairall, C. W., Yang, M., Bariteau, L., Edson, J. B., Helmig, D., McGillis, W., Pezoa, S., Hare, J. E., Huebert, B., and Blomquist, B.: Implementation of the Coupled Ocean-Atmosphere Response Experiment flux algorithm with CO2, dimethyl sulfide, and O3, J. Geophys. Res.: Oceans, 116, n/a–n/a, doi:10.1029/2010JC006884, 2011.

Fredriksson, S. T., Arneborg, L., Nilsson, H., and Handler, R. A.: Surface shear stress dependence of gas transfer velocity parameterizations using DNS, Journal of Geophysical Research: Oceans, 121, 7369–7389, doi:10.1002/2016JC011852, http://dx.doi.org/10.1002/2016JC011852, 2016.

Frew, N. M., Bock, E. J., Schimpf, U., Hara, T., Haußecker, H., Edson, J. B., McGillis, W. R.,

Nelson, R. K., McKenna, S. P., Uz, B. M., and Jähne, B.: Air-sea gas transfer: Its dependence on wind stress, small-scale roughness, and surface films, J. Geophys. Res., 109, C08S17, 2004.

Jähne, B., Huber, W., Dutzi, A., Wais, T., and Ilmberger, J.: Wind/wave-tunnel experiment on the Schmidt number—And wave field dependence of air/water gas exchange, in: Gas transfer at water surfaces, edited by Brutsaert, W. and Jirka, G. H., pp. 303–309, D. Reidel, 1984.

Jähne, B., Münnich, K. O., Bösinger, R., Dutzi, A., Huber, W., and Libner, P.: On the parameters influencing air-water gas exchange, J. Geophys. Res.: Oceans, 92, 1937–1949, doi:10.1029/JC092iC02p01937, http://dx.doi.org/10.1029/JC092iC02p01937, 1987.

Krall, K. E.: Laboratory Investigations of Air-Sea Gas Transfer under a Wide Range of Water Surface Conditions, PhD. Ruperto-Carola University of Heidelberg, Germany, 2013.

McKenna, S. and McGillis, W.: The role of free-surface turbulence and surfactants in air–water gas transfer, International Journal of Heat and Mass Transfer, 47, 539 – 553, doi:https://doi.org/10.1016/j.ijheatmasstransfer.2003.06.001, http://www.sciencedirect.com/science/article/pii/S001793100300423X, 2004.

Soloviev, A. V.: Coupled renewal model of ocean viscous sublayer, thermal skin effect and interfacial gas transfer velocity, Journal of Marine Systems, 66, 19 – 27, doi:http://dx.doi.org/10.1016/j.jmarsys.2006.03.024, http://www.sciencedirect.com/science/article/pii/S0924796306001709, 2007.

Wilczak, J., Oncley, S., and Stage, S.: Sonic Anemometer Tilt Correction Algorithms, Boundary-Layer Meteorology, 99, 127–150, doi:10.1023/A:1018966204465, 2001.

Woolf, D. K.: Bubbles and their role in gas exchange, in: The Sea Surface and Global Change, edited by Liss, P. S. and Duce, R. A., pp. 173–206, Cambridge University Press, http://dx.doi.org/10.1017/CBO9780511525025.007, cambridge Books Online, 1997.

Zhang, Q., Handler, R. A., and Fredriksson, S. T.: Direct numerical simulation of turbulent free convection in the presence of a surfactant, International Journal of Heat and Mass Transfer, 61, 82 – 93, doi:https://doi.org/10.1016/j.ijheatmasstransfer.2013.01.031, http://www.sciencedirect.com/science/article/pii/S0017931013000549, 2013.

---

## Author Comment (AC2) · 29 Dec 2017

**Response to Referee $\#2$**

**RC** : *This paper presents a new corrections of the estimated air-sea CO2 gas flux in ship measurements considering the air-flow distortion. And the air-sea CO2 gas transfer velocity using this corrections showed the smaller variation than the former result. This manuscript contains possibly interesting points for the readers of Atmospheric Chemistry and Physics. However, the Result and Discussion part is not described adequately. The following are the concerns and some suggestions.*

[Figure]

**AC** : We would like to thank the reviewer for his constructive comments and suggestion. Please find our responses below.

**RC** : *In Section 3.2, the authors mentioned the residual flow distortion error about the disagreement with COARE 3.5 parameterization. Consequently, the authors should describe the reasons in detail.*

**AC** : We added the following to Section 3.2:
"For the wind speeds these can be (i) errors in the estimated acceleration/deceleration of the relative wind speed; (ii) errors in the estimated horizontal deflection, which will lead to minor inaccuracies in the correction for horizontal ship velocity; and (iii) errors in the estimated uplift, which would introduce bias in the wind speed normalisation. For the friction velocities, bias in estimates can arise from (i) insufficient removal of the ship-motion signals (MSC) and (NAV); (ii) small inaccuracies in the tilt estimate; and (iii) uncertainties in the estimation of the elevated cospectral energy for $n \geq 1\,Hz$."

**RC** : *In Section 3.3, The authors should clearly mention about the accuracy of catamaran's data. For example, what is the method of the motion correction for the catamaran's data? And how about the accuracy of the catamaran's wind speed?*

**AC** :  Thanks for the hint, we added the following information to the section 3.3:
"During periods of fair weather, wind speed and direction were also measured by an Airmar PB200 marine sonic anemometer at 5.6 m a.s.l. on the mast of a small catamaran. The PB200 has an RMS uncertainty of $0.5\,m\,s^{-1}$ at wind speeds $< 5\,m\,s^{-1}$, which increases to $1\,m\,s^{-1}$ for higher wind speeds. A GPS incorporated in the unit was used to correct the measured speeds for horizontal platform motion."

**RC** : *In Section 4.1, since the CO2 gas transfer velocity decrease in the high wind speed in Fig. 8, it is better to add the applicable wind speed range to Eq. (10).*

**AC** : The observed decrease at high wind speeds is based on only 4 samples from a single high wind speed event. More measurements at wind speeds above $20\,m\,s^{-1}$ will be necessary to
accurately predict gas exchange at these extreme conditions. Based on SOAP, we suggest that Eq. (10) is applicable to the wind speed range $5 - 19\,\mathrm{m\,s^{-1}}$. Please also refer to response to Referee #1 for more details.

---

## Author Comment (AC4) · 10 Jan 2018

**Response to Referee #3**

**RC** : *This paper is a careful reanalysis of the eddy covariance measurements of momentum and carbon dioxide fluxes from the Southern Ocean Surface Ocean Aerosol Production study (SOAP). Every aspect of motion correction and flow distortion on the means and fluxes is considered and appropriate corrections applied. The smooth dependence on wind speed, of the friction and mass transfer velocities, suggests that the corrections are accurate and complete. The slightly higher friction velocities than those from COARE 3.5 may be a fetch effect: COARE was open ocean while SOAP was shorter fetch corresponding to higher drag coefficients. The divergence from quadratic of the CO2 mass transfer velocity above wind speed of 16 m/s has important repercussions for the global carbon budget and is the main scientific (rather than technical) product of this fine work.*

**AC** : We would like to thank the reviewer for this very positive review. The fetch was variable during SOAP, but the predominant wave direction was from the south-south-west (Law et al., 2017, Fig. 6) which has a long fetch. The full set of wind back-trajectories is available if needed. But a few examples using a particle tracking model can be found in (Law et al., 2017, Fig. 5). These show that for southerly wind direction, the fetch was actually considerable. For the SOAP experiment which was conducted east of New Zealand wind direction can be used as a proxy for fetch, with northerly and westerly wind directions being related to limited fetch, while southerly and easterly wind directions feature open ocean fetch conditions. A map of the cruise track can be found in (Law et al., 2017, Fig. 1).

In Fig. 1 the observed neutral drag coefficient $C_{\mathrm{D10N}}$ is shown as function of $u_{10\mathrm{N}}$, which was computed from the air flow distortion corrected wind speed measurements from the bow mast. The data are also shown as $1\,\mathrm{m\,s^{-1}}$ wind speed bin-averages, which were computed for the total data set and for the four quadrants of the true wind direction northerly, easterly, southerly, and westerly, which here denote true wind directions between $315°$ and $45°$, $45°$ and $135°$, $135°$ and $225°$, and $225°$ and $315°$, respectively. No dependence of the drag coefficient is visible for wind speeds above $7\,\mathrm{m\,s^{-1}}$. For lower wind speeds the drag coefficient appears to have some dependence on the true wind direction.

The exercise is repeated in Fig. 2, using the relative wind direction to the bow instead of the true wind direction. Fig. 2 reveals a much stronger dependence of the measured $C_{\mathrm{D10N}}$ on the relative wind direction, which is visible for the full range of observed wind speeds and is likely caused by residual flow distortion bias either in the $u_*$ or in the $u_{10\mathrm{N}}$ estimates. For lower wind speeds the contribution of the ship's speed over ground to the relative wind speed becomes relatively more important. This can lead to relatively higher errors in $u_{10\mathrm{N}}$ and may explain the higher variability in the observed $C_{\mathrm{D10N}}$. Notably the relative wind direction sector $-22.5°$ to $+22.5°$ shows the least variability of $C_{\mathrm{D10N}}$ at low wind speeds. This sector is dominated by station measurements, since the ships bow was pointed into the prevailing wind direction during the stations, whenever possible. Based on these observations we suggest that during SOAP $C_{\mathrm{D10N}}$ was biased higher than COARE 3.5 due to residual air flow distortion effects in $u_{10\mathrm{N}}$ to a greater degree than any fetch effects. We have added the following sentence to Sec. 3.3 *Effect of the air flow distortion corrections on friction velocity*: "The neutral drag coefficient $C_{\mathrm{D10N}} = u_*^2\,u_{10\mathrm{N}}^{-2}$ computed from the measurements showed no dependence on the true wind direction, which could have indicated an effect of the varying fetch. The measured $C_{\mathrm{D10N}}$ varied, however, on average by about $\pm 7\%$ with the relative wind direction for relative wind directions within $\pm 90°$ to the bow."

[Figure]

Figure 1: Neutral drag coefficient $C_{D10N}$ as function of $u_{10N}$. The air flow distortion corrected wind speed measurements from the bow mast were used to calculate $u_{10N}$ and $C_{D10N} = (u_* u_{10N}^{-1})^2$. The data are shown as individual measurements and bin averaged over $1\,\mathrm{m\,s^{-1}}$ wind speed bins, which are calculated for the total data set and the four quadrants of true wind directions (e.g. easterly means wind directions between $45°$ and $135°$). The the COARE 3.5 drag coefficient (Edson et al., 2013) is shown for comparison. Bin average values consist of five or more individual measurements. The errorbars indicate the standard error of the mean value of each wind speed and direction bin.

[Figure]

Figure 2: Same as Fig. 1, but the data are grouped by the wind direction relative to the bow of the ship. Here $\alpha = 0$ indicates that the ship was pointed into the prevailing wind direction.

**References**

45  Edson, J. B., Jampana, V., Weller, R. A., Bigorre, S. P., Plueddemann, A. J., Fairall, C. W., Miller, S. D., Mahrt, L., Vickers, D., and Hersbach, H.: On the Exchange of Momentum over the Open Ocean, J. Phys. Oceanogr., 43, 1589–1610, doi:10.1175/JPO-D-12-0173.1, 2013.

Law, C. S., Smith, M. J., Harvey, M. J., Bell, T. G., Cravigan, L. T., Elliott, F. C., Lawson, S. J., Lizotte, M., Marriner, A., McGregor, J., Ristovski, Z., Safi, K. A., Saltzman, E. S., Vaattovaara,
50  P., and Walker, C. F.: Overview and preliminary results of the Surface Ocean Aerosol Production (SOAP) campaign, Atmospheric Chemistry and Physics, 17, 13 645–13 667, doi:10.5194/acp-17-13645-2017, URL https://www.atmos-chem-phys.net/17/13645/2017/, 2017.